# A bifunctional snoRNA with separable activities in guiding rRNA 2′-O-methylation and scaffolding gametogenesis effectors

Estelle Leroy[1,5], Drice Challal[1,4,5], Stéphane Pelletier [1], Coralie Goncalves[2], Alexandra Menant[1], Virginie Marchand [3], Yan Jaszczyszyn[1], Erwin van Dijk[1], Delphine Naquin [1], Jessica Andreani [1], Yuri Motorin [3], Benoit Palancade [2] & Mathieu Rougemaille [1] ✉

Small nucleolar RNAs are non-coding transcripts that guide chemical modifications of RNA substrates and modulate gene expression at the epigenetic and post-transcriptional levels. However, the extent of their regulatory potential and the underlying molecular mechanisms remain poorly understood. Here, we identify a conserved, previously unannotated intronic C/D-box snoRNA, termed *snR107*, hosted in the fission yeast long non-coding RNA *mamRNA* and carrying two independent cellular functions. On the one hand, *snR107* guides site-specific 25S rRNA 2′-O-methylation and promotes pre-rRNA processing and 60S subunit biogenesis. On the other hand, *snR107* associates with the gametogenic RNA-binding proteins Mmi1 and Mei2, mediating their reciprocal inhibition and restricting meiotic gene expression during sexual differentiation. Both functions require distinct *cis*-motifs within *snR107*, including a conserved 2′-O-methylation guiding sequence. Together, our results position *snR107* as a dual regulator of rRNA modification and gametogenesis effectors, expanding our vision on the non-canonical functions exerted by snoRNAs in cell fate decisions.

Conserved throughout the eukaryotic kingdom, small nucleolar RNAs (snoRNAs) constitute a class of non-coding transcripts that guide chemical modifications of cognate RNA substrates through sequence complementarity[1,2]. The two main H/ACA and C/D-box families of snoRNAs are characterized by distinctive *cis*-motifs and secondary structures, and assemble with dedicated enzymes and cofactors to catalyze pseudouridylation or ribose 2′-O-methylation, respectively[1–3]. Produced from intergenic transcripts or hosted within introns of pre-mRNAs or long non-coding RNAs (lncRNAs), mature snoRNAs target various classes of substrates, including ribosomal RNAs (rRNAs), transfer RNAs (tRNAs), small nuclear RNAs (snRNAs) as well as mRNAs[1,4,5]. snoRNA-guided modifications or base-pairing directly impact RNA processing, folding and/or stability, thereby contributing to major cellular processes such as ribosome biogenesis and activity, mRNA translatability and pre-mRNA splicing[2,4]. Beyond these canonical functions, snoRNAs also modulate chromatin accessibility and can provide a source of piRNA/miRNA-like sRNAs to control histone modifications and RNA degradation[2]. SnoRNAs thus exhibit broad roles in the control of gene expression and their link to various human diseases, including cancer, suggests that these functions contribute to cell fate decisions[1,2,6].

Gametogenesis is a fundamental developmental process required for the transmission of the genetic information to offspring in all sexually reproducing species. In the fission yeast *Schizosaccharomyces*

[1]Université Paris-Saclay, CEA, CNRS, Institute for Integrative Biology of the Cell (I2BC), 91198 Gif-sur-Yvette, France. [2]Université Paris Cité, CNRS, Institut Jacques Monod, F-75013 Paris, France. [3]Université de Lorraine, Epitranscriptomics and RNA sequencing (EpiRNA-Seq) Core Facility (SMP IBSLor) and UMR7365 IMoPA CNRS, Nancy, France. [4]Present address: Expression Génétique Microbienne, UMR8261 CNRS, Université Paris Cité, Institut de Biologie Physico-Chimique, 75005 Paris, France. [5]These authors contributed equally: Estelle Leroy, Drice Challal. ✉e-mail: mathieu.rougemaille@i2bc.paris-saclay.fr

*pombe*, gamete production depends on the successful completion of meiosis, which initiates upon integration of intra- and extracellular signals and triggers the sequential induction of hundreds of genes[7,8]. In mitotic cells, a group of meiotic genes is also constitutively transcribed but the corresponding mRNAs are selectively eliminated by an RNA degradation system involving the YTH-family RNA-binding protein Mmi1, thereby avoiding untimely expression[9]. Mmi1 binds to a specific *cis*-motif, known as DSR (*i.e.* Determinant of Selective Removal) that is enriched in repeats of the hexanucleotide UNAAAC, and targets meiotic transcripts to the nuclear exosome for rapid degradation[9–14]. Upon sexual differentiation, this RNA silencing pathway is turned off by a ribonucleoprotein (RNP) complex, composed of the meiosis-specific RNA-binding protein Mei2 and the DSR-containing lncRNA *meiRNA* decoy, that sequesters and inactivates Mmi1 at the *meiRNA*-encoding *sme2+* locus, thus ensuring efficient meiosis progression[9,15–17].

Selective elimination of DSR-containing meiotic mRNAs in mitotic cells requires multiple factors, among which the small nuclear protein Erh1 that assembles with Mmi1 to form the EMC heterotetramer (Erh1-Mmi1 Complex)[18–20], components of the polyadenylation/termination machinery[21–23] and the multisubunit MTREC/NURS complex that physically bridges Mmi1 to the nuclear exosome[24–29]. In mitotic cells, these complexes promote the deposition of repressive heterochromatin marks on meiotic genes and localize to scattered nuclear foci to promote retention and degradation of meiotic mRNAs[9,18,19,22–25,27,29–33]. In parallel, Mmi1 associates with the Ccr4-Not complex and the lncRNA *mamRNA* to target its own meiotic inhibitor Mei2 for ubiquitinylation and downregulation, thereby preserving its activity in the degradation of meiotic transcripts[34–37]. Colocalizing in one of the Mmi1-enriched nuclear foci, *mamRNA* not only restricts Mei2 accumulation but also promotes Mmi1 inactivation by high Mei2 levels in the absence of the E3 ubiquitin ligase subunit Mot2 of Ccr4-Not, therefore holding at the center of the Mmi1-Mei2 mutual control in mitotic cells[37]. However, the *cis*-elements of *mamRNA* required for the binding and regulation of both RNA-binding proteins and their contribution to meiotic gene expression during sexual differentiation have remained unknown.

Based on a combination of genomic, imaging and biochemical approaches, we report here that *mamRNA* encodes a conserved, previously unannotated intronic C/D-box snoRNA, termed *snR107*, that not only guides site-specific 25S rRNA 2'-O-methylation but also mediates the regulatory control of Mmi1 and Mei2 activities in both proliferative and meiotic cells. Our results thus reveal an unanticipated interplay between small and long ncRNAs in the control of gametogenesis and expand our understanding of the non-canonical functions exerted by snoRNAs.

## Results

### *mamRNA* is produced from a *U14*-containing precursor and undergoes splicing

Transcribed upstream of the *U14* snoRNA-encoding gene (Fig. 1a), *mamRNA* accumulates as two non-adenylated isoforms and a pool of polyadenylated species[37]. To further investigate its biogenesis and processing, we conducted RNA-seq experiments and compare total, ribo-depleted RNA to oligo(dT)-enriched, polyadenylated transcripts. Consistent with former observations[37], *mamRNA* was enriched in both fractions, whereas mature *U14* accumulated preferentially in the total fraction, in agreement with snoRNAs being non-adenylated transcripts (Fig. 1a). Remarkably, the coverage of a 227 nucleotide-long window located upstream of *U14* was severely reduced in the poly(A)+ fraction specifically. Visual inspection of the corresponding sequence revealed the presence of canonical splicing sites, strongly suggesting the presence of an intron (Fig. 1a).

To determine whether *mamRNA* is produced from a *U14*-containing precursor transcript undergoing splicing, we performed semi-quantitative RT-PCR experiments using primers surrounding the

intron, *i.e.* in the 5' region of *mamRNA* and in *U14* (Fig. 1b). In wild type cells, two main products were detected, the faster-migrating spliced species being more abundant, in agreement with RNA-seq data (Fig. 1a). Strikingly, mutations of the 5' splice site (5'SS), the branch-point sequence (BP) and the 3' splice site (3'SS), alone or in combination, led to a massive accumulation of the unspliced transcript (Fig. 1b). Additional products were also detected in the 5'SS and 3'SS mutants, supporting the existence of alternative splicing signals.

We further characterized the RNA species by Northern blotting (NB) using four probes covering distinct parts of the locus. Probe A targeting the 5' region of *mamRNA* detected two main isoforms in wild type cells, consistent with our former analyses[37], while probe D revealed the mature form of *U14* (Fig. 1c). Interestingly, probe B unveiled a stable intronic RNA product that migrated slower upon deletion of the debranching enzyme Dbr1, suggestive of a splicing-dependent lariat (Fig. 1c, Supplementary Fig. 1a). In 5'&3'SS$_{mut}$ and BP&3'SS$_{mut}$ cells, all four probes detected high levels of a longer RNA species (Fig. 1c, Supplementary Fig. 1b), which ended a few hundred nucleotides downstream of *U14* as determined by 3' RACE analyses (Supplementary Fig. 1c). Of note, the mature form of *U14* was barely affected in these cells, indicating that splicing contributes only marginally, if at all, to its biogenesis (Fig. 1c). Together, our data indicate that a long *mamRNA-U14* precursor is efficiently spliced and processed in wild type cells to generate *mamRNA* isoforms (probe A), an intronic product (probe B) and *U14* (probe D). The massive increase in steady state levels of the intron-retaining RNA further suggests that the transcript is targeted for decay following splicing.

We previously showed that one of the Mmi1 nuclear foci colocalizes with the *mamRNA* transcription site[37]. The presence of an intron at the locus prompted us to re-examine this spatial overlap depending on the occurrence of splicing. Using a set of probes specific to the 5' exon of *mamRNA* (*i.e.* nucleotides 11 to 510 from TSS) in single molecule RNA FISH experiments (smFISH), we found that Mmi1-*mamRNA* colocalization persisted in the absence of the intron or upon defective splicing (Fig. 1d). Interestingly, 5'&3'SS$_{mut}$ cells also exhibited a strong increase in signal intensity and a redistribution of the lncRNA to the nucleolus, as determined by fluorescence overlap with the nucleolar marker Nop56 (Fig. 1d, e). Together, our data indicate that defective splicing not only impairs *mamRNA* turn-over but also confers its nucleolar localization, likely through retention of the intron and/or *U14*.

### *mamRNA* intron encodes a C/D-box snoRNA

To delineate the species produced from the *mamRNA-U14* locus in wild type cells, we conducted direct RNA sequencing (dRNA-seq) using Oxford Nanopore Technology (ONT) (Fig. 2a). We obtained a total of 26,497,493 reads from two sequencing runs with a median read length of ~230 nt. Consistent with the above results, a few reads covered the entire locus, demonstrating the existence of a precursor transcript encompassing both *mamRNA* and *U14*. Together with mature, 5' and/or 3'-extended *U14* species, spliced transcripts initiating at the *mamRNA* TSS and ending just upstream or downstream of the intron were also detected (Fig. 2a, lower panel), presumably corresponding to the two isoforms observed by NB (Fig. 1c, probe A; Supplementary Fig. 1a, probe C).

dRNA-seq further revealed the accumulation of a short species embedded in the intron (Fig. 2a, lower panel), consistent with NB data (Fig. 1c, probe B). 5' and 3' RACE analyses indicated that the RNA is 107 nt long and inspection of its sequence revealed the presence of canonical motifs distinctive of C/D-box snoRNAs that are involved in RNA 2'-O-methylation (2'-O-Me)[38–40] (Supplementary Fig. 2a). These include the highly conserved C and D boxes (UGAUGA and CUGA respectively) as well as the more degenerate C' and D' boxes (UGACGA and CCUA respectively) (Fig. 2b, Supplementary Fig. 2a). Based on these features, we named this previously unannotated transcript *snR107* (*s*mall *n*ucleolar *R*NA *107*). Multiple sequence alignment

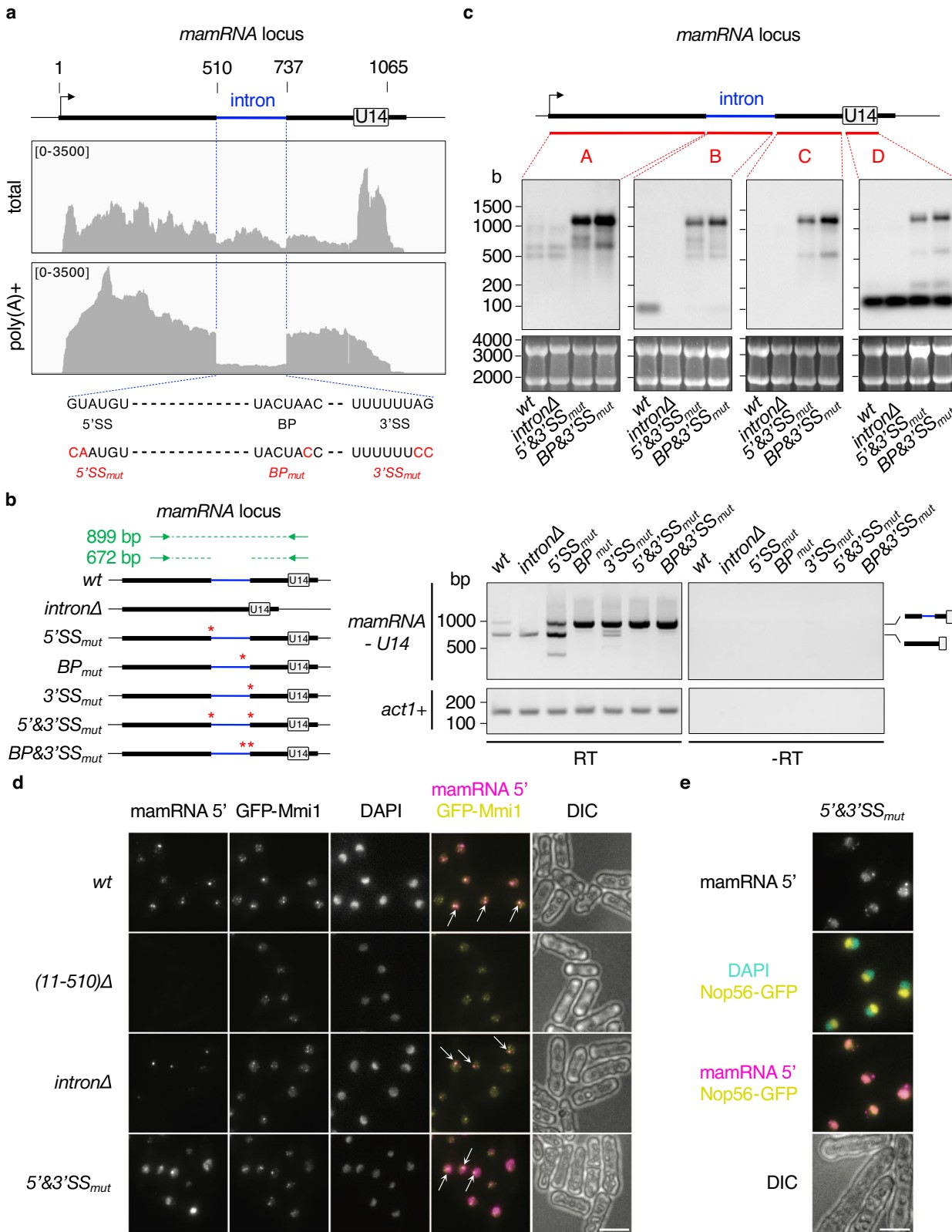

indicated that *snR107* is conserved in eukaryotes, with orthologues in budding yeast (*snR190*) and humans (*SNORD12/B/C*) (Fig. 2b). Beyond the C/D and C'/D' boxes, sequence homology extended to two motifs lying upstream of the D' and D boxes, hereafter referred as ASE1 and ASE2 (for AntiSense Element 1 and 2), that are complementary to sequences in the 25S/28S rRNA (Fig. 2b, Supplementary Fig. 2b). Remarkably, *S. pombe snR107* and *S. cerevisiae snR190* also showed

syntenic organization, both being encoded upstream of the highly conserved *U14* snoRNA[41]. Yet, *snR190* is processed by endonucleolytic cleavage[41], while human *SNORD12/B/C* locate in consecutive introns of the lncRNA ZFAS1[42] (Supplementary Fig. 2c).

To assess whether *snR107* encodes a bona fide C/D-box snoRNA, we performed RNA-immunoprecipitation coupled to Northern blotting (RIP-NB) experiments using core snoRNP components as bait

**Fig. 1 | *mamRNA* is produced from a *U14*-containing precursor and undergoes splicing. a** Total and poly(A) + RNA-seq profiles of the *mamRNA* and *U14*-encoding genomic locus (n = 2). The numbers above the scheme indicate the nucleotide length from *mamRNA* TSS (black arrow). The numbers in square brackets indicate the range of reads. Below are shown the sequences at the borders of the intron with the 5′ splice site (5′SS), branchpoint (BP) and 3′ splice site (3′SS) highlighted in bold. Nucleotides mutated in dedicated strains are shown below in red. **b** Semi-quantitative RT-PCR assay of the *mamRNA-U14* locus. *Left*, scheme showing the different strains used, with point mutations labeled with red stars. The primers used are denoted by green arrows. *Right*, cDNA analysis on agarose gels. *act1+* was used as a loading control and no RT reactions (-RT) were loaded in parallel. bp = base pairs. **c** Northern blots showing *mamRNA* and *U14* levels from total RNA samples, in the indicated genetic backgrounds. The position of the different probes used (A, B, C and D) are depicted on top. Ribosomal RNAs served as loading control (lower panels). b = bases. **d** and **e** Representative images of *mamRNA* localization in mitotic cells of the indicated genetic backgrounds, expressing either GFP-tagged Mmi1 or Nop56. smFISH probes used were specific of *mamRNA* 5′ exon and DNA was stained with DAPI. Images are shown as Z-projections. Scale bar, 5 μm. In **d** white arrows point to Mmi1 dot-*mamRNA* colocalization. Source data are provided as a Source Data file.

proteins, including the 2′-O-methyltransferase Fib1 and its partners Nop56, Nop58 and Snu13[38,39]. All four factors efficiently immunoprecipitated *snR107*, as well as the C/D-box *U14* snoRNA (Fig. 2c). Note that Nop58 also associated with a longer species, presumably the full intron, suggestive of co-transcriptional binding and stepwise snoRNP assembly. We further carried out smFISH analyses and found that *snR107*, as opposed to *mamRNA* 5′ exon, accumulated specifically in the nucleolus in wild type cells (Fig. 2d, e). The signal was specific since the same set of probes did not detect any signal upon deletion of *snR107* (Fig. 2d), in which the biogenesis of *mamRNA* and *U14* was largely preserved (Supplementary Fig. 2d). Cells lacking the 5′ exon of *mamRNA* (*i.e.* (11-510)Δ) did not alter the localization of *snR107*, although the latter was partially retained in a longer *U14*-containing transcript (Fig. 2d, Supplementary Fig. 2d). In 5′&3′SS_mut cells, whereby *snR107* is trapped within the unspliced lncRNA, the nucleolar localization was also observed (Fig. 2d), consistent with the above results (Fig. 1e). Together, our results demonstrate that *snR107* is a canonical C/D-box snoRNA encoded in *mamRNA* intron.

The 5′ and 3′ ends of intronic snoRNAs are generally processed through the action of dedicated 3′-5′ and 5′-3′ exoribonucleases[1]. We found that the absence of the nuclear exosome subunit Rrp6 and of the TRAMP poly(A) polymerase Cid14 accumulated the mature form of *snR107*, suggesting a role for these factors in downregulating the snoRNA (Supplementary Fig. 2e). Oppositely, manipulating the levels of *mamRNA*-bound RNA-binding proteins such as Mmi1 (*mmi1Δ*) or Mei2[37], which accumulates in the absence of the E3 ubiquitin ligase subunit of Ccr4-Not (*mot2Δ*), did not alter *snR107* levels (Supplementary Fig. 1a, probe B). As for 5′-3′ exoribonucleases, depletion of Dhp1, but not deletion of Xrn1, also led to the accumulation of longer species (Supplementary Fig. 2e, f), implying that the former contributes to the processing of *snR107* 5′ end. Thus, while Dhp1 participates in *snR107* processing, TRAMP and Rrp6 are involved in its turnover.

### *snR107* guides 25S rRNA 2′-O-methylation and is required for ribosome biogenesis

C/D-box snoRNAs guide site-specific RNA 2′-O-Me through sequence complementarity[38-40]. Previous liquid chromatography coupled to mass spectrometry analyses identified the 2′-O-methylated guanosine 2483 (Gm2483) in the *S. pombe* 25S rRNA that lies in a region complementary to the *snR107* ASE1[43] (Supplementary Fig. 3a). To determine whether *snR107* mediates G2483 2′-O-Me, we performed RiboMethSeq experiments, an Illumina sequencing-based method that detects 2′-O-methylated ribonucleotides as gaps in 5′- and 3′-end coverage due to RNA protection from alkaline fragmentation (Fig. 3a)[44,45]. Remarkably, G2483 2′-O-Me was highly prevalent in the wild type strain but absent in cells lacking *snR107*, expressing a mutated version of the ASE1 (snR107_ASE1mut) or defective for *mamRNA* splicing (5′&3′SS_mut) (Fig. 3a, Supplementary Data 1). This effect was specific, as neither the second region of the 25S rRNA (residues 333 to 343) potentially targeted by *snR107* ASE2 nor other highly modified nucleotides exhibited *snR107*-dependent 2′-O-Me (Supplementary Fig. 3a, b, Supplementary Data 1). Thus, splicing-released *snR107* guides Gm2483 deposition via the ASE1.

Global analysis of RiboMethSeq data for the 5S, 5.8S, 18S and 25S rRNA sequences revealed 0, 2, 39 and 53 high confidence 2′-O-methylated residues (i.e. RNA Methylation Scores > 0.8) respectively, most of which were previously detected by mass spectrometry[43], confirming the reliability of the method (Supplementary Data 1). Interestingly, although a large fraction of 2′-O-Me sites is conserved in budding yeast[46] (Supplementary Data 1), the corresponding G2483 in *S. cerevisiae* (i.e. G2395) is not modified due to the lack of *snR190* 2′-O-Me guiding activity[47]. *S. pombe snR107* therefore evolved rRNA modification properties, akin to its human counterpart *SNORD12C*[48].

To get further insights into the mechanism of Gm2483 deposition, we generated additional *snR107* mutants, including the reverse complement sequence of a conserved 6 nt motif between the D′ and C′ boxes that is part of the kink-loop (snR107_kloopmut)[3], the deletion of 21 nt downstream of the C′ box (snR107_21IntΔ) predicted to fold as a stem-loop structure, and the reverse complement sequence of the ASE2 upstream of the D box (snR107_ASE2mut) (Fig. 3b, Supplementary Fig. 2b). We carried out an RNAseH-based assay relying on the enzymatic cleavage of a hybrid formed by the 25S rRNA and a chimeric RNA/DNA oligonucleotide whose DNA moiety targets the G2483 residue[49] (Fig. 3c, left panel). In this assay, the presence of 2′-O-Me at the targeted position prevents hybrid cleavage by RNAseH. Addition of the chimeric oligonucleotide in the presence of the enzyme specifically generated two 25S rRNA cleavage products of about 2.5 and 1.0 kb in the snR107Δ, snR107_ASE1mut and 5′&3′SS_mut strains (Fig. 3c, see blue and red arrows in SYBR-stained gel), consistent with a loss of Gm2483. NB experiments using a probe directed against the 3′ part of 25S rRNA confirmed that these mutants accumulated cleavage products compared to wild type cells (Fig. 3c, see red arrow in NB panel). Interestingly, we found that formation of Gm2483 was also abolished in snR107_kloopmut, while its level was only marginally or partially impaired in snR107_21IntΔ and snR107_ASE2mut respectively (Fig. 3c). Hence, in addition to the dedicated guiding sequence, other motifs within *snR107* are important for efficient Gm2483 deposition.

We next sought to investigate whether *snR107* contributes to rRNA biogenesis, beyond Gm2483 modification. We found that the 35S rRNA precursor accumulates to various levels in snR107Δ, snR107_kloopmut, snR107_ASE2mut and 5′&3′SS_mut cells but not in the snR107_ASE1mut strain, supporting a role for the snoRNA in pre-rRNA processing that is distinguishable from its function as a guide for Gm2483 deposition (Supplementary Fig. 3c).

To further evaluate the functional impact of *snR107* activities on ribosomes, we analyzed polysome profiles from cell extracts on sucrose gradients. snR107Δ, snR107_kloopmut, snR107_ASE2mut and 5′&3′SS_mut cells all exhibited reduced levels of free 60S and 80S particles relative to 40S subunits as well as half-mer polysomes, indicative of a lower pool in 60S subunits and/or translation defects (Fig. 3d). The patterns of the snR107_ASE1mut and snR107_21IntΔ strains were instead identical to wild type cells, demonstrating that Gm2483 and the putative stem-loop structure are not critical determinants of 60S subunit biogenesis (Fig. 3d, Supplementary Fig. 3d). Importantly, the different phenotypes observed in *snR107* mutants were not due to altered RNA accumulation (Supplementary Fig. 3e) nor to impaired nucleolar localization (Supplementary

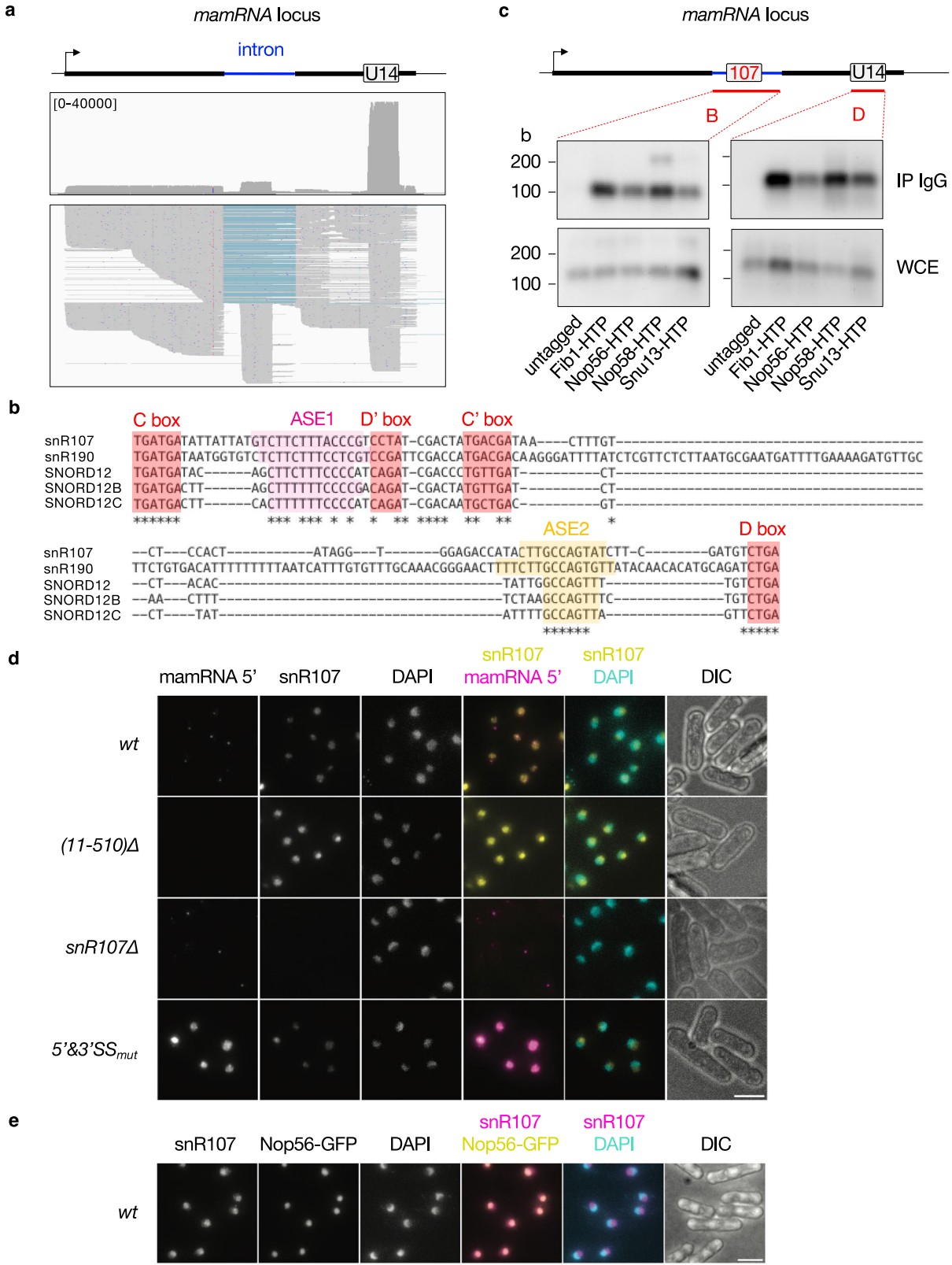

Fig. 3f), and they were not associated with substantial growth defects (Supplementary Fig. 3g). From these experiments, we concluded that *snR107* mediates 25S rRNA $G^{2483}$ 2′-O-Me and contributes to pre-rRNA processing and 60S subunit biogenesis in a $Gm^{2483}$-independent fashion. These two functions in ribosome biogenesis require distinct antisense elements (ASE1 and ASE2, respectively), suggesting that they both involve base-pairing with rRNA.

### *snR107* mediates the Mmi1-Mei2 mutual control

We previously showed that Mmi1 associates with *mamRNA* to target its own inhibitor Mei2 for downregulation by the Ccr4-Not complex[37]. Consistently, *MMI1* inactivation or near-complete deletion of *mamRNA* (*i.e.* (11-737)Δ) both led to increased levels of TAP-tagged Mei2, as assessed by Western blotting (Fig. 4a, b). To investigate the contribution of *snR107* to this regulation, we constructed

**Fig. 2 | mamRNA intron encodes a C/D-box snoRNA. a** dRNA-seq profile of the *mamRNA* and *U14*-encoding genomic locus. The top box shows the cumulative coverage (n = 2), with the range of reads indicated in square brackets. The bottom box shows a subset of reads (200) representative of the different read populations. The following down-sampling parameters in IGV were used: sampling window size (bases) = 230 (*i.e.* median read length); numbers of reads per window = 200. The blue bars correspond to intronic sequences that are absent in the corresponding reads. **b** Multiple sequence alignment (CLUSTALW) of *S. pombe snR107*, *S. cerevisiae snR190* and human *SNORD12/B/C*. The C, D′, C′ and D boxes are highlighted in red. The antisense elements (ASE1 and ASE2) complementary to the 25S or 28S rRNAs are highlighted in magenta and orange respectively. Conserved residues are denoted with an asterisk. Nucleotides immediately upstream of the C box and downstream of the D box were excluded for the analysis. **c** Northern blots showing the fractions of *snR107* and *U14* immunoprecipitated with HTP-tagged versions of Fib1, Nop56, Nop58 and Snu13. An untagged strain was used as negative control. IP = Immunoprecipitate; WCE = Whole Cell Extract. b = bases. **d, e** Representative images of *mamRNA* and *snR107* localization as detected by smFISH in cells of the indicated genetic backgrounds. DNA was stained with DAPI. Images are shown as Z-projections. Scale bar, 5 μm. Source data are provided as a Source Data file.

several deletion mutants of the *mamRNA* precursor (Fig. 4a). Remarkably, while the absence of the 5′ exon alone (*i.e.* (11-510)Δ) did not impact Mei2 abundance, deletion of the intron or *snR107* resulted in increased protein levels (Fig. 4b). Expression of a plasmid-borne version of *mamRNA* intron allowed the production of mature *snR107* species (Supplementary Fig. 4a) and further rescued low Mei2 levels in otherwise mamRNA$_{(11-737)Δ}$ cells (Fig. 4c), demonstrating that *snR107* is necessary and sufficient for this function. *mamRNA*-hosted *snR107* is therefore the key determinant involved in Mei2 downregulation.

We next analyzed cells defective for *mamRNA* splicing. Although individual inactivation of splice sites (5′SS$_{mut}$, BP$_{mut}$, 3′SS$_{mut}$) only slightly impacted, if at all, Mei2 levels, the protein accumulated in the 5′&3′SS$_{mut}$ and BP&3′SS$_{mut}$ strains similar to *snR107Δ* cells (Fig. 4d). The relative steady state levels of mature *snR107* in splicing-defective *mamRNA* mutants most likely account for these differences, as the free snoRNA was more severely reduced in the 5′&3′SS$_{mut}$ and BP&3′SS$_{mut}$ strains compared to individual splice site mutants (Supplementary Fig. 4b). We also did not observe increased Mei2 levels in the absence of Dbr1 (Supplementary Fig. 4c), indicating that *snR107* is still functional for Mei2 downregulation when trapped in a lariat (Supplementary Fig. 1c, probe B). To determine the impact of specific *cis*-elements within *snR107*, we further assessed Mei2 abundance in the *snR107* mutants mentioned above. While snR107$_{ASE1mut}$ and snR107$_{21IntΔ}$ cells maintained low Mei2 levels, the protein accumulated in the snR107$_{kloopmut}$ and snR107$_{ASE2mut}$ strains (Fig. 4e), revealing the requirement for specific motifs. Importantly, these results indicate that the ASE1 motif responsible for 25S rRNA G$^{2483}$ 2′-O-Me does not partake in the control of Mei2 levels, thereby uncoupling both *snr107* activities.

To exclude the possibility that Mei2 accumulates as a consequence of defective ribosome activity in relevant *snR107* mutants (Figs. 3d, 4e), we analyzed cells lacking Cgr1 (Supplementary Fig. 4d), a non-essential factor previously involved in pre-rRNA processing and biogenesis of pre-60S ribosomal particles in budding yeast[50,51]. Importantly, while the absence of Cgr1 resulted in a strong reduction in the levels of monosomes (80S) and the appearance of pronounced half-mers (Supplementary Fig. 4e), Mei2 levels remained low in this context (Supplementary Fig. 4f). We concluded that the accumulation of Mei2 in the corresponding *snR107* mutants is not caused by indirect effects in ribosome biogenesis and/or activity.

Upon Mei2 accumulation in the absence of the Ccr4-Not ubiquitin ligase Mot2, Mmi1 is inactivated in a *mamRNA*-dependent manner, resulting in increased steady state levels of meiotic mRNAs in mitotic cells[37]. We investigated the role of *snR107* in this process and found that its absence was sufficient to suppress the accumulation of *mei4* +, *ssm4* +, *mcp5* + mRNAs and *meiRNA* in a *mot2Δ* background, phenocopying *mamRNAΔ*[37] (Fig. 4f). Remarkably, while mutation of *snR107* ASE1 had no effect, *mot2Δ* snR107$_{kloopmut}$ cells showed only marginal increase in meiotic transcripts, indicative of a major requirement for the corresponding motif in the Mmi1-Mei2 mutual control (Fig. 4f). Expression of snR107$_{ASE2mut}$ or splicing-defective *mamRNA* (5′&3′SS$_{mut}$) also resulted in the decrease of meiotic mRNAs, albeit to a lesser extent (Fig. 4f). Moreover, ectopic expression of *mamRNA*

intron in *mot2Δ* mamRNA$_{(11-737)Δ}$ cells was sufficient to restore increased levels of meiotic transcripts (Supplementary Fig. 4g), implying that Mmi1 inactivation by high Mei2 levels solely depends on *snR107*. Together, our data demonstrate the prominent role for splicing-released *snR107* in the regulation of Mmi1 and Mei2 activities in vegetative cells and further unveil the importance of specific *cis*-acting motifs, distinguishable from the ASE1 element guiding G$^{2483}$ 2′-O-Me.

## snR107 restricts the amplitude and timing of meiotic gene expression

In *S. pombe*, meiosis onset critically depends on inactivation of the Pat1 kinase, allowing in turn Mei2 activation and hence Mmi1 inhibition[9,52]. To assess the role of *snR107* during sexual differentiation, we induced synchronized meiosis using an ATP analogue-sensitive mutant of Pat1 (*pat1-L95G*) that can be chemically inactivated from nitrogen-starved, G1-arrested haploid cells[53] (Fig. 5a). Following addition of the Pat1 inhibitor 3-MB-PP1, we harvested cells at different time points (Supplementary Fig. 5a) and assessed the expression profiles of *snR107* and *meiRNA*, the Mmi1 lncRNA decoy. *meiRNA* strongly accumulated between 1 and 5 h after Pat1 inactivation (Fig. 5b), which corresponds to the premeiotic phase before completion of meiosis I (Supplementary Fig. 5a), in agreement with the reported *meiRNA* accumulation at the horse-tail stage of meiosis in diploid cells[17,20,37]. This dynamic expression profile was not observed for *snR107*, whose levels remain largely similar throughout meiosis, with the exception of a moderate increase in nitrogen-starved cells (Fig. 5b).

We next determined the expression dynamics of the *mcp5* + and *ssm4* + meiotic mRNAs that are targeted for degradation by Mmi1 during mitosis. In wild type cells, both transcripts exhibited a sharp increase in expression levels, reaching a maximum 3 hours after Pat1 inactivation, prior to a rapid decline the following hours (Fig. 5c). This pattern resembled that of *meiRNA*, suggesting that meiotic mRNAs accumulate as a consequence of Mmi1 sequestration. Interestingly, mamRNA$_{(11-510)Δ}$ cells only marginally affected the kinetics of *mcp5* + and *ssm4* + expression (Fig. 5c). Conversely, the absence of *snR107* radically impacted the profiles of both transcripts, which were induced earlier, stronger and over a much longer time window, up to 6-7 hours post Pat1 inactivation (Fig. 5c). These data indicate that *snR107*, but not the 5′ exon of *mamRNA*, restricts the timing and amplitude of Mmi1-dependent meiotic gene expression during sexual differentiation. The minor changes observed in cells lacking the ribosome biogenesis factor Cgr1 (Supplementary Fig. 5b) further support the notion that altered expression profiles in *snR107* mutants accumulating Mei2 are not the mere consequence of defective ribosome biogenesis and/or reduced translational efficiency, but most likely due to changes in Mmi1 and/or Mei2 activities.

Following the same strategy, we subsequently determined the impact of *mamRNA* splicing sites and *snR107 cis*-acting motifs on meiotic gene expression. Mirroring the effect on Mei2 levels in vegetative cells (Fig. 4d), inactivation of the 3′ splice site (3′SS$_{mut}$) alone barely affected *mcp5* + and *ssm4* + mRNA expression profiles, whereas 5′&3′SS$_{mut}$ cells showed a broader range of induction (Fig. 5d).

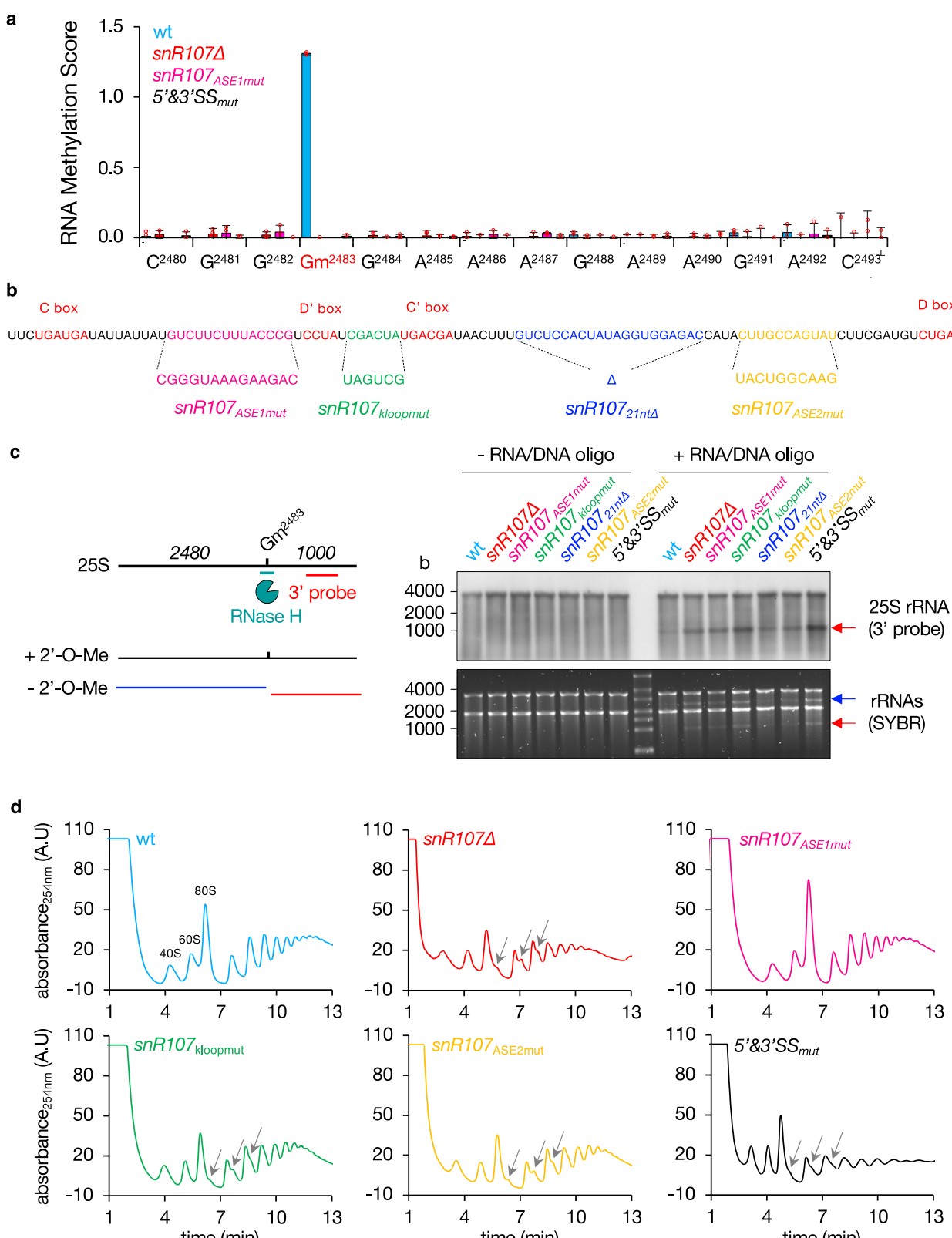

Likewise, snR107_ASE1mut cells behaved identically to the wild type, but both snR107_kloopmut and snR107_ASE2mut strains exhibited higher and wider expression of *mcp5*+ and *ssm4*+ mRNAs, the former almost phenocopying *snR107Δ* cells (Fig. 5e). To confirm these findings, we expanded our gene-specific analyses genome-wide by performing RNA-seq experiments in wild type and snR107_kloopmut cells. Similarly, most genes belonging to the Mmi1 regulon reached their maximal

expression 3 to 4 hours following addition of 3-MB-PP1 in wild type cells, while they tend to accumulate to higher levels and for a longer period in the snR107_kloopmut strain, a pattern that was specific to Mmi1 targets[10,19] (Fig. 5f, Supplementary Fig. 5c). These experiments together demonstrate that the same *snR107* motifs involved in Mei2 down-regulation in vegetative cells also restrict meiotic gene expression during meiosis. Importantly, the observed effects could not be solely

**Fig. 3 | *snR107* guides 25S rRNA 2'-O-methylation and is required for ribosome biogenesis. a** RNA methylation scores (RMS) of the 25S rRNA residues 2480 to 2493 (complementary to *snR107* ASE1) as determined by RiboMethSeq in strains of the indicated genotypes (mean$_{(RMS-median\ all\ RMS)}$ ± SD; n = 3 biological replicates). Individual data points are represented by red circles. **b** Scheme depicting the *snR107* sequence with color-coded motifs mutated in strains of interest (snR107$_{ASE1mut}$, snR107$_{kloopmut}$, snR107$_{21Int\Delta}$, snR107$_{ASE2mut}$). The C, D', C' and D boxes are labeled in bold and red. **c** RNAseH-based cleavage assay. *Left*, Scheme depicting the 25S rRNA with the positions of Gm$^{2483}$, the RNA/DNA oligonucleotide used for RNAseH cleavage (green) and the Northern probe (red). Below are shown the 25S rRNA products obtained with or without 2'-O-Me. *Right*, Northern blot showing 25S rRNA in the presence or absence of the RNA/DNA oligonucleotide, from total RNA samples in the indicated genetic backgrounds. Ribosomal RNAs served as loading control. The blue and red arrows denote the 5' and 3' cleavage products, respectively. b = bases. **d** Polysome profiles from total cellular extracts obtained on sucrose gradients in strains of the indicated genetic backgrounds. Absorbance at 254 nm was measured for up to 13 min of collection time and is expressed as arbitrary units (A.U). The gray arrows point to half-mers. Source data are provided as a Source Data file.

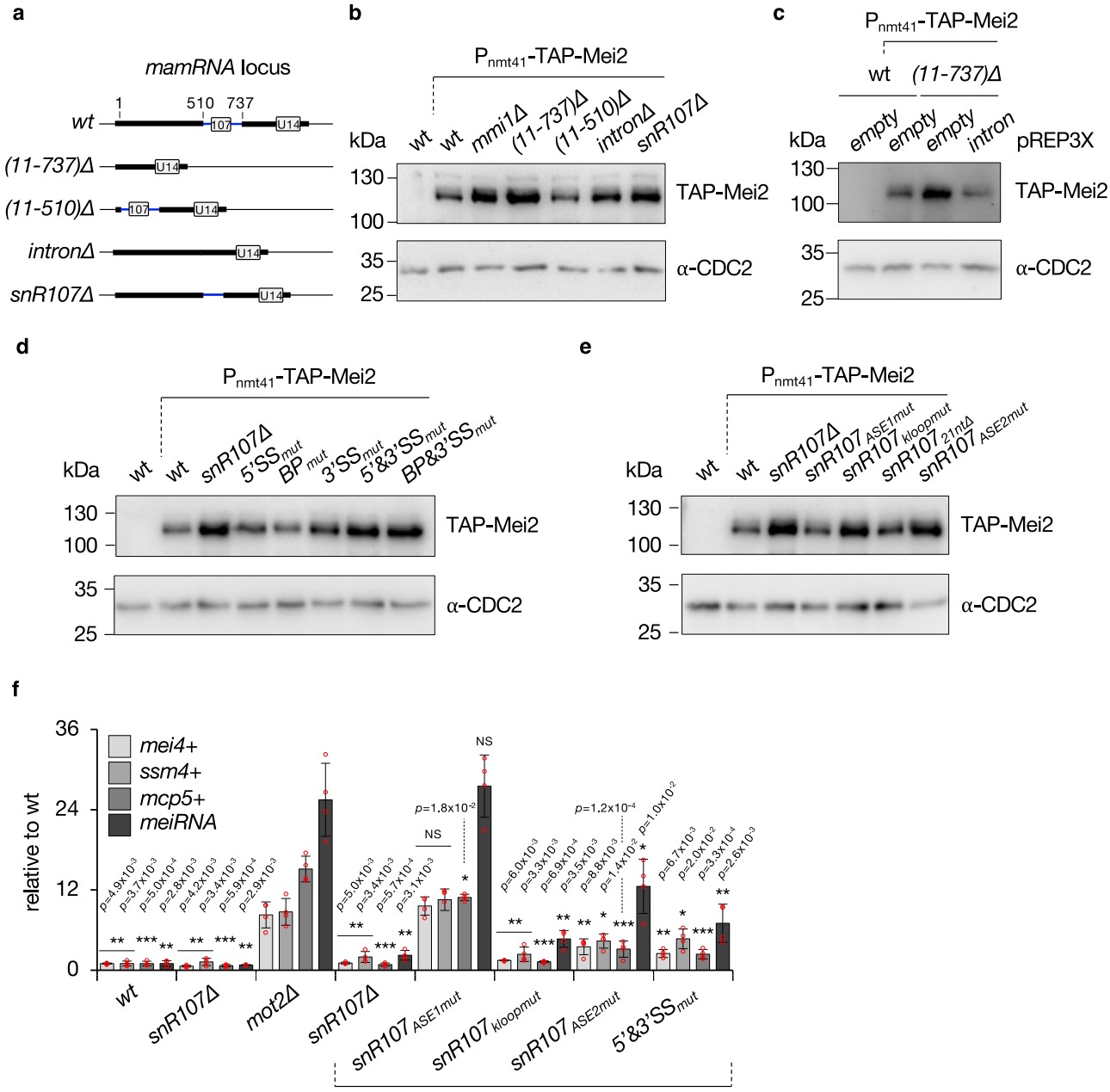

**Fig. 4 | *snR107* mediates the Mmi1-Mei2 mutual control. a** Scheme depicting the different deletion mutants of the *mamRNA-U14* locus. **b** to **e** Western blots showing the levels of TAP-tagged Mei2 in cells of the indicated genetic backgrounds. An anti-CDC2 antibody was used as loading control. kDa = kilodaltons. **f** RT-qPCR analyses of *mei4+*, *ssm4+*, *mcp5+* meiotic mRNA and *meiRNA* levels in cells of the indicated genetic backgrounds (mean ± SD; n = 4 biological replicates; normalized to *act1+* and relative to wt). Student's t-test (two-tailed) was used to calculate p-values (relative to *mot2Δ*). Individual data points are represented by red circles. NS = not significant. Source data are provided as a Source Data file.

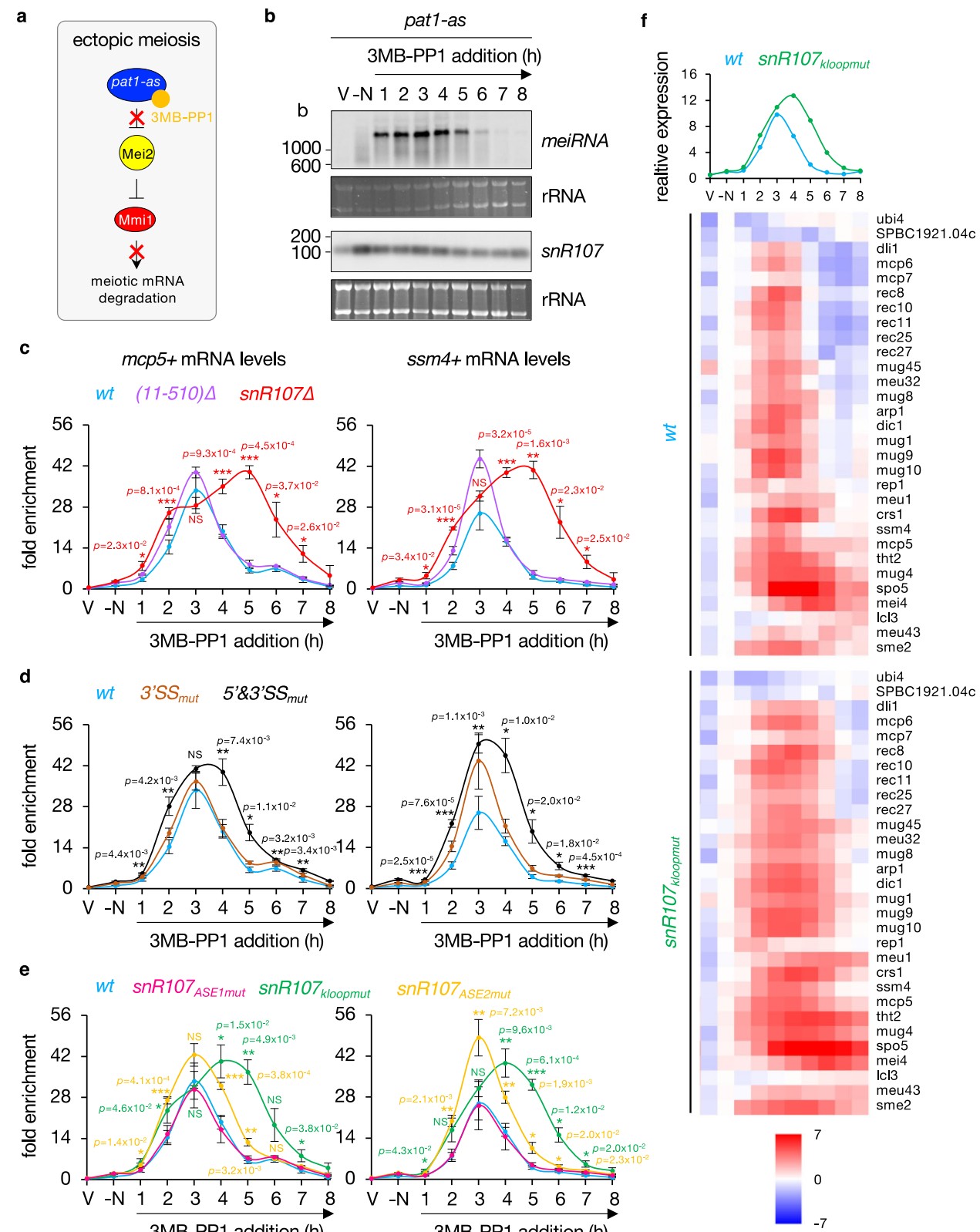

attributed to a delay in the progression of haploid meiosis, as all *mamRNA* and *snR107* mutants slowed down meiosis to a similar extent compared to wild type cells (Supplementary Fig. 5d), irrespective of their impact on meiotic gene expression.

During the horse-tail stage of meiotic prophase, Mmi1 is sequestered by its inhibitors Mei2 and *meiRNA* within a unique nuclear dot overlapping *meiRNA* transcription site[16,17]. Using smFISH analyses, we found that *snR107* accumulates in several foci, none of which colocalized with *meiRNA* and Mmi1 (Supplementary Fig. 5e). Likewise, the absence of *snR107* or expression of splicing-defective *mamRNA* (5'&3'SS_mut) did not impact sporulation efficiency, as opposed to *meiRNAΔ* cells (Supplementary Fig. 5f). Our results therefore support the notion that *snR107* limits Mmi1 inhibition during meiosis, likely by hindering excessive Mei2 activity.

**Fig. 5 | *snR107* restricts the amplitude and timing of meiotic gene expression. a** Scheme depicting the induction of ectopic meiosis by inhibition of the *pat1-as* allele (*pat1-L95G*) with 3-MB-PP1, which in turn activates the Mmi1 inhibitor Mei2 and hence prevents meiotic mRNA degradation. **b** Northern blots showing *meiRNA* and *snR107* levels from total RNA samples isolated at different time points following addition of 3-MB-PP1 in wild type cells. Ribosomal RNAs served as loading control. V = vegetative cells; -N = nitrogen-starved cells. b = bases. **c** to **e** RT-qPCR analyses of *mcp5+* and *ssm4+* meiotic mRNA levels upon induction of meiosis in cells of the indicated genetic backgrounds (mean ± SD; n = 3 biological replicates

for (*11-510*)Δ, *snR107Δ*, *3'SS_{mut}*, *5'&3'SS_{mut}*, *snR107_{kloopmut}* and *snR107_{ASE2mut}* strains; n = 4 biological replicates for *wt* and *snR107_{ASE1mut}* strains; normalized to total RNA concentration and relative to wt -N). Note that data for the wild type strain in **c** were replotted in **d** and **e** to ease comparison. Student's t-test (two-tailed) was used to calculate p-values (relative to wt). NS = not significant. **f** Poly(A) + RNA-seq analyses of wild type and *snR107_{kloopmut}* cells upon meiosis induction (n = 2). Shown are the median expression profiles of the Mmi1 regulon[10] and the corresponding heatmaps for each individual gene (relative to wt -N; log2 scale). V = vegetative cells; -N = nitrogen-starved cells. Source data are provided as a Source Data file.

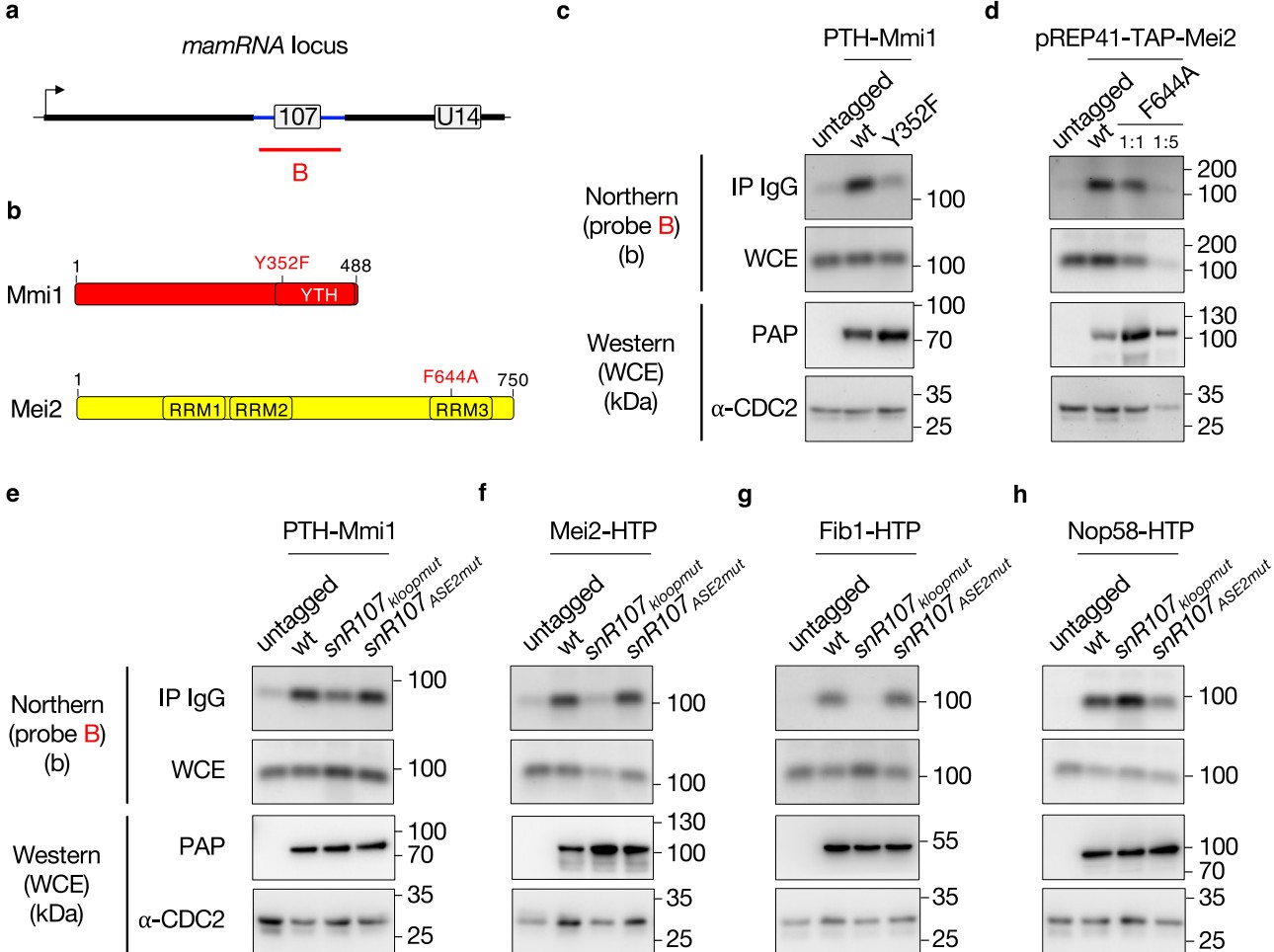

**Fig. 6 | Mmi1 and Mei2 associate with *snR107* and the 2'-O-methyltransferase Fib1. a** Scheme representing the *mamRNA* locus with the probe B used for Northern blotting analyses in **c** to **h**. **b** Scheme depicting the domain organization of Mmi1 and Mei2. **c** to **h** *Upper panels*: Northern blots showing the fractions of *snR107* immunoprecipitated in cells of the indicated genetic backgrounds expressing HTP or TAP-tagged versions of Mmi1 (**c** and **e**), Mei2 (**d** and **f**), Fib1 (**g**) or Nop58 (**h**). Untagged strains were used as negative controls. In **d**, plasmid-borne, TAP-tagged versions of wild type and mutated Mei2 (F644A) were used.

1:5 dilutions of IP and WCE were loaded to assess the association of the Mei2 mutated version, as compared to its wild type counterpart. In **f**, Mei2 immunoprecipitation was carried out 3 h post Pat1 inactivation. (b) = bases. *Lower panels*: Western blots showing total levels of HTP or TAP-tagged Mmi1, Mei2, Fib1, and Nop58, detected with PAP in the corresponding whole cell extracts (WCE). An anti-CDC2 antibody was used as a loading control. IP = Immunoprecipitate; (kDa) = kilodaltons. Source data are provided as a Source Data file.

## Mmi1 and Mei2 associate with *snR107* and the 2'-O-methyltransferase Fib1

We had initially identified *mamRNA* as simultaneously bound by Mmi1 and Mei2 in sequential RNA immunoprecipitation experiments[37]. Since the *snR107* moiety is sufficient to account for the functional impact of *mamRNA* on Mmi1 and Mei2 activities, we anticipated that both proteins should associate with *snR107*. We performed RIP-NB experiments and found that Mmi1 and Mei2 indeed interact with *snR107* (Fig. 6a–d, compare *untagged* and *wt*). Given the importance of their RNA-binding activities in their mutual

control in vegetative cells[37], we also assessed their association with *snR107* upon mutations of their respective YTH and RRM3 domains (Fig. 6b). Both Mmi1-Y352F and Mei2-F644A, which are defective for the binding to *mamRNA* 5' exon[37], showed severely reduced interactions (Fig. 6c, d, for Mei2, compare *wt* to *F644A 1:5* for equivalent bait protein levels), indicating that the proteins bind to *snR107* through their RNA-binding domains. Moreover, Mmi1 did not promote the recruitment of Fib1 to the snoRNA nor compete with it for efficient binding, as revealed by the Mmi1-independent association of the 2'-O-methyltransferase (Supplementary Fig. 6a).

a

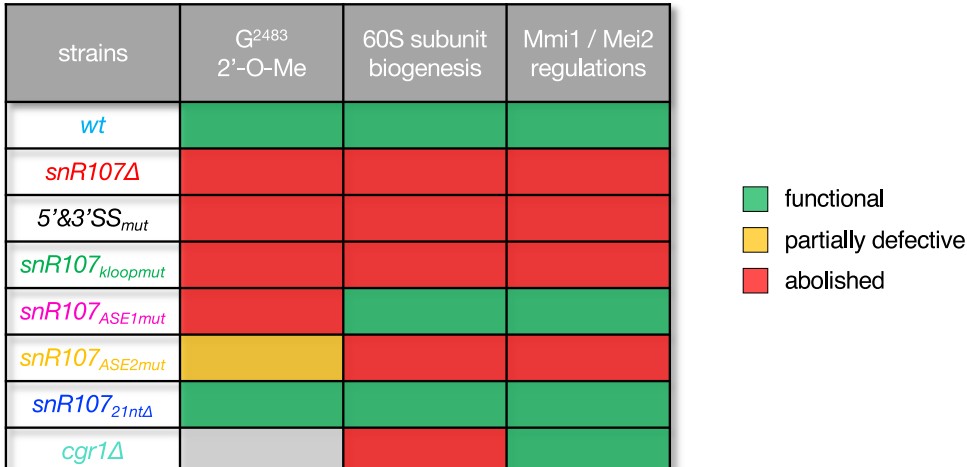

| strains | G$^{2483}$ 2'-O-Me | 60S subunit biogenesis | Mmi1 / Mei2 regulations |
|---|---|---|---|
| *wt* | functional | functional | functional |
| *snR107Δ* | abolished | abolished | abolished |
| *5'&3'SS$_{mut}$* | abolished | abolished | abolished |
| *snR107$_{kloopmut}$* | abolished | abolished | abolished |
| *snR107$_{ASE1mut}$* | abolished | functional | functional |
| *snR107$_{ASE2mut}$* | partially defective | functional | functional |
| *snR107$_{21ntΔ}$* | functional | functional | functional |
| *cgr1Δ* | undetermined | abolished | functional |

- functional (green)
- partially defective (orange)
- abolished (red)

b

25S rRNA 2'-O-Me                                   Mmi1 / Mei2 regulations

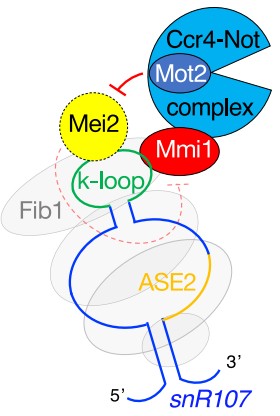

**Fig. 7 | Summary and model of *snR107* functions. a** Table summarizing the requirements for *snR107* cis-acting motifs and Cgr1 in rRNA 2'-O-Me, 60S subunit biogenesis and the regulations of Mmi1 and Mei2. Green, orange and red boxes refer to functional, partially defective and abolished activities, respectively. The gray box indicates undetermined. **b** Model depicting the roles of *snR107* in rRNA 2'-O-Me and the control of Mmi1 and Mei2 activities. Through its ASE1, *snR107* guides 25S rRNA G$^{2483}$ 2'-O-Me by Fib1 (left). It further contributes to pre-rRNA processing and 60S subunit biogenesis in a Gm$^{2483}$-independent manner (not shown). *snr107* also mediates the Mmi1-Mei2 mutual control in vegetative cells and restricts meiotic gene expression during sexual differentiation in an ASE2-dependent manner (right). The k-loop motif is critical for both regulations by promoting the binding of Mmi1, Mei2, and Fib1. Other snoRNP components (Nop56, Nop58, Snu13) are represented as gray circles.

The requirement of the k-loop and ASE2 sequences for the Mmi1-Mei2 mutual control and meiotic gene expression prompted us to assess their contribution to the association of both proteins with *snR107*. Strikingly, expression of snR107$_{kloopmut}$ resulted in a reduced or even complete loss of association of Mmi1, Mei2 and Fib1, but not Nop58 (Fig. 6e–h). Importantly, the lack of Fib1 interaction was specific, as judged by its persistent association with *U14* in this background (Supplementary Fig. 6b). An opposite trend was observed in snR107$_{ASE2mut}$ cells, whereby only Nop58 showed a partially decreased interaction (Fig. 6e–h). Together, these results indicate that the k-loop motif contributes to anchor Mmi1, Mei2 but also Fib1 to *snR107*.

To further characterize the modalities of Mmi1 and Mei2 interactions with the *snR107*-containing snoRNP, we investigated whether both factors physically associate with Fib1 using in vivo co-immunoprecipitation assays. Such interactions were detected in wild type, snR107$_{kloopmut}$ and snR107$_{ASE2mut}$ cells (Supplementary Fig. 6c, d), indicating that Mmi1 and Mei2 bind to Fib1, and presumably other core snoRNP factors, independently of their association with

*snR107*. In line with the possibility that both proteins interact with additional C/D-box snoRNAs, we detected their association with U14 as well, consistent with previous genome-wide analyses[12,37,54] (Supplementary Fig. 6e, f). Thus, Mmi1 and Mei2 bind to distinct snoRNA-containing RNPs, *snR107* being critical for the regulation of their activities.

## Discussion

Our study shows that the lncRNA *mamRNA* undergoes splicing and hosts a previously unannotated, intronic C/D-box snoRNA, termed *snR107*, involved in both 25S rRNA 2'-O-Me / 60S subunit biogenesis and the regulation of the gametogenic RNA-binding proteins Mmi1 and Mei2 (Fig. 7).

Upon defective *mamRNA* splicing, *snR107* and downstream *U14* are trapped within a stable precursor transcript. Nonetheless, the levels of mature *U14* are not significantly impacted, suggesting that it may also be produced from an independent promoter, presumably downstream of the *snR107*-containing intron. Regardless, the

massively increased levels of the intron-retaining species imply a tight coupling between RNA splicing and decay. The RNAseIII-family endoribonuclease Pac1, which downregulates non-coding precursor transcripts[55], including a 5′-extended *U14* isoform, may not operate upon retention of *mamRNA* intron, thereby stabilizing the unspliced product. Interestingly, the production of mature *snR190* and *U14* in *S. cerevisiae* solely relies on Pac1 homolog Rnt1-mediated endonucleolytic cleavage of a single precursor transcript[41,56]. Fission yeast therefore evolved splicing signals to generate *snR107*, which recalls human *SNORD12/B/C* that are hosted in introns of the lncRNA *ZFAS1*[42]. Beyond these maturation steps, *snR107* biogenesis also requires the recruitment of snoRNP components such as Fib1 and Nop58. *snR107* RNP assembly could initiate during transcription, as supported by the scored association of Nop58 with *mamRNA* intron and by the localization of Mmi1 at the *mamRNA* locus. Dissecting the coupling between *snR107* transcription, processing, and assembly with protein partners, including Mmi1 and Mei2, will be the focus of future investigation.

The involvement of *mamRNA* splicing-released *snR107* in controlling Mmi1 and Mei2 activities points to spatially-confined regulations in the nucleolus, which is critical for various nuclear functions and cell fate decisions[57,58]. Beyond its canonical role in ribosome biogenesis, the nucleolus coordinates responses to multiple stresses to in turn adapt cell physiology and destiny[57,58]. This relies on the activation of signaling pathways that modulate rDNA transcription / rRNA processing and affect protein trafficking to and out of the nucleolus for regulatory purposes[57,58]. The *snR107*-dependent reciprocal inhibitions of Mmi1 and Mei2 in vegetative cells may reflect a similar principle. On the one hand, *snR107* may couple efficient 25S rRNA 2′-O-Me and processing to the Mmi1 and Ccr4-Not-dependent maintenance of low Mei2 levels in the nucleolus, thereby sustaining robust mitotic growth. On the other hand, Mei2 accumulation in the absence of Mot2 may impair nuclear degradation of meiotic mRNAs by precluding Mmi1 release from the nucleolus and/or by diffusing in the nucleoplasm. Regardless, the kinetics of snoRNP assembly/disassembly and/or the dynamics of protein shuttling are likely to provide an additional layer in fine-tuning Mmi1 and Mei2 activities to adjust meiotic gene expression in response to environmental changes. Whether such regulations may also involve additional snoRNA partners requires further investigation.

Sexual differentiation, which triggers the accumulation of Mei2 and *meiRNA*[15,17,59], initiates upon nutritional starvation and is accompanied by a massive decrease in cellular rRNA content. Such nucleolar stress may directly contribute to the progression of meiosis, possibly releasing a substantial amount of Mei2 in the nucleoplasm to associate with and sequester Mmi1 at the *meiRNA* transcription site[17]. The increased and persistent accumulation of meiotic mRNAs upon defective association of Mmi1 and Mei2 with *snR107* likely reflects the loss of a buffering system that normally prevents excessive Mmi1 inhibition to allow timely meiotic gene expression and facilitate the sequence of events leading to sporulation. In this scenario, a fraction of Mmi1 may still interact with *snR107*, and possibly other mRNA targets, supporting a fine balance between protective and inhibitory mechanisms of Mmi1 activity during meiosis. Our results thus provide an example of a developmentally-regulated switch whereby an RNA-binding protein (Mmi1) cooperates with a snoRNA (*snR107*) to buffer the sponging activity of its own lncRNA decoy (*meiRNA*) by limiting the abundance and/or the activity of the associated partner (Mei2).

In parallel to the regulation of Mmi1 and Mei2 activities, *snR107* guides 25S rRNA G$^{2483}$ 2′-O-Me via its ASE1, which is, however dispensable for pre-rRNA processing and 60S subunit biogenesis. Located in the root helix 73 of the 25S rRNA domain V, in proximity to the ribosome Peptidyl Transferase Center (PTC)[43], Gm$^{2483}$ may be functionally relevant only under given circumstances and the observed delayed meiosis in snR107$_{ASE1mut}$ cells is intriguing in this respect. Alternatively, the loss of 2′-O-Me may be compensated by structural

rearrangements, as previously seen for mutations of conserved residues in the PTC[60]. Regardless of the contribution of Gm$^{2483}$, pre-rRNA processing and 60S subunit biogenesis require integrity of *snR107* ASE2, whose complementary sequence in 25S rRNA is devoid of modification[43]. Akin to *S. cerevisiae snR190* that functions as a chaperone rather than a guide in the maturation of pre-60S ribosomal particles[47,61,62], ASE2 probably base-pairs with pre-rRNA to promote 2′-O-Me-independent folding and/or remodeling early during processing. How this could in turn impact optimal Gm$^{2483}$ deposition as well as ribosome activity requires further investigation. Likewise, whether the human orthologues *SNORD12/B/C*, which mediate ASE1-dependent 28S rRNA G$^{3878}$ 2′-O-Me[48,63] and possess relatively small ASE2, also partake in these processes remains to be established.

Our analyses revealed that *snR107* ASE1 does not contribute to the control of Mmi1 and Mei2 activities, thereby uncoupling 25S rRNA G$^{2483}$ 2′-O-Me from Mei2 downregulation and meiotic gene expression. Mutations of the k-loop or ASE2 instead impair these different functions, with lower Gm$^{2483}$ levels most likely attributed to the loss or suboptimal binding of Fib1 and Nop58, respectively. Yet, only the k-loop is important for efficient interactions of Mmi1 and Mei2, which suggests that i) the control of their activity requires a fully assembled snoRNP, with ASE2 ensuring the correct recruitment and positioning of Nop58 and possibly other core components, and ii) they recognize, together with Fib1, the same sequence element within *snR107*. Given the spacing constraints imposed by the size of the motif, an attractive scenario may lie in the formation of a dimeric snoRNP[64], whereby two protein-bridged *snR107* molecules would bring two k-loops in close proximity. This could ensure the simultaneous binding of Mmi1, Mei2 and Fib1 to fulfill their respective functions. Intriguingly, although additional sequences beyond the sole k-loop are likely to participate in the recruitment of Mmi1, the fact that *snR107* lacks the typical YTH-bound UNAAAC motifs might suggest the existence of an alternative mode of RNA-binding. The absence within the k-loop motif of consecutive Us, which are bound by Mei2 RRM3 in vitro[37,65], could similarly reflect a specific mode of RNA recognition. Alternatively, it is possible that Mmi1 and/or Mei2 are engaged in protein-protein interactions with snoRNP components through their RNA-binding domains. This could involve Fib1 itself and/or Snu13, which directly recognizes the kink-turn conformation within C/D-box snoRNAs[1,64]. Defining the precise mode of snoRNP assembly and the possible conservation of such macromolecular entities in other systems is a challenge for future studies.

Over the last years, snoRNAs have gained strong interest due to their misexpression in various human diseases, including cancers and developmental disorders[6]. However, this link is merely correlative in most cases, and it remains unclear whether additional, non-canonical functions may rationalize these physiological defects. The recently-defined role of human SNORA13 in promoting cellular senescence independently of its pseudouridylation guiding activity has illuminated the functional versatility of snoRNAs in cell fate decisions[66]. Our findings that *snR107* tunes the activity of gametogenesis effectors further add to the regulatory potential of snoRNAs and provide compelling evidence for a conserved and ancestral role in developmental transitions. The existence of similar mechanisms in humans may help understanding the etiology of diseases and defining therapeutic strategies.

## Methods

### Strains and plasmids

The *Schizosaccharomyces pombe* strains used in this study are listed in Supplementary Table 1 and were generated by lithium acetate-based transformation using complete medium (YE Broth, Formedium, #PMC0105) supplemented with appropriate antibiotics (G418, hygromycin B, nourseothricin). PCR fragments used for epitope tagging or gene deletion were obtained from genomic DNA or plasmids (Supplementary Table 2).

To generate strains expressing mutated versions of *mamRNA* or *snR107*, the former was first deleted with a cassette containing the hygromycin B resistance marker (hph$^R$MX) fused to the herpes simplex virus thymidine kinase-encoding gene (HSV-TK) from the pFA6a-HyTkAX vector (Addgene plasmid #73898; RRID:Addgene_73898)[67]. The resulting hph$^R$ and TK-expressing strains were then transformed with a PCR product carrying the desired mutations. Integrants were selected in the presence of 5-fluoro-2′-desoxyuridine, which counter-selects cells expressing TK, and sensitivity to hygromycin B due to cassette pop-out was systematically confirmed. Positive clones were verified by diagnostic PCR and Sanger sequencing. A similar strategy was followed to construct strains expressing wild type or mutated versions of N-terminally tagged Mmi1 from its own promoter, except that initial deletion of the ORF with the hph$^R$TK cassette was performed in *mei4Δ* cells to overcome viability defects due to ectopic expression of the meiosis-specific transcription factor Mei4[9].

For vector-based expression of *snR107*, the *mamRNA* intron was PCR-amplified and cloned in the multicopy pREP3X plasmid (between the SalI and BamHI sites) under the control of the strong nmt1 promoter. Following transformation, cells carrying the resulting pREP3X-mamRNA$_{intron}$ vector were selected on minimal medium lacking leucine (EMM-LEU).

## Media and growth conditions

All experiments were performed with mid-log phase cells grown at 30 °C in EMM-LEU-URA minimal medium (EMM-LEU-URA Broth, Formedium, #PMD0810) supplemented with 150 mg/L of L-leucine and uracil (EMM) or EMM-LEU-URA supplemented with uracil (EMM-LEU) for plasmids-containing strains.

For growth assays, exponentially growing cells were plated on EMM at an initial OD = 0.2 followed by 5-fold serial dilutions. Pictures were taken after 3 days of growth at 30 °C.

Induction of haploid meiosis was performed as previously described[53]. Briefly, heterothallic h- strains carrying the *pat1-L95G* allele were grown in EMM at 30 °C prior to overnight (15 to 16 h) incubation in EMM lacking nitrogen (EMM-N) (EMM Broth without nitrogen, Formedium, #PMD1305). The following day, cultures were supplemented with 50 mg/L L-leucine, 500 μg/mL NH$_4$Cl, and 25 μM 3-MB-PP1 (Sigma, #529582) to induce synchronized meiosis and aliquots were harvested every h for up to 8-9 h.

To assess mating/sporulation efficiency, exponentially growing homothallic h90 cells were plated on ME plates (ME Broth, Formedium, #PCM0710) for 3 days at 30 °C, prior to exposition to iodine crystals, which stain with dark color a starch-like compound in the spore wall.

For auxin-induced degradation of degron-tagged proteins (i.e., Dhp1$^{3mAID-5FLAG}$ expressing cells), 100% ethanol-solubilized indole-3-acetic acid was added to the culture medium (5 mM final concentration) for 2 h before harvesting cells.

## Total RNA extraction

Total RNAs were extracted from 4 to 8 mL of yeast cells grown in exponential phase. Cell pellets were resuspended in 400 μL TES buffer (10 mM Tris-HCl pH7.5, 5 mM EDTA, 1% SDS) and 400 μL acid phenol solution pH 4.3 (Sigma, #P4682), prior to incubation at 65 °C for 1 h in a Thermomixer (20 sec ON/OFF, 1400 rpm). After centrifugation, the aqueous phase was mixed with 400 μL chloroform (Acros Organics, #383760010). Samples were centrifuged again and ethanol precipitated in the presence of 0.2 M lithium chloride. A total of 15 μg glycogen (ThermoFisher Scientific, #R0561) were further added to RNA samples prepared upon meiosis induction. Pellets were then washed in 70% ethanol, resuspended in water and treated with DNase (Ambion, #AM1906). RNA concentrations were measured with Nano-Drop or DeNovix DS-11 devices. Total RNA samples obtained following meiosis induction were aligned at 400 ng/μL prior to RT-qPCR or Northern blotting analyses.

## Semi-quantitative RT-PCR and RT-qPCR

Reverse transcription reactions were performed as previously described[37], except random hexamers (ThermoFisher Scientific, #SO142) and oligodT were used for cDNA synthesis.

For semi-quantitative RT-PCR analyses, 2 μL of diluted cDNAs were amplified with Dream Taq DNA polymerase (ThermoFisher Scientific, #EP0713). For *mamRNA-U14* cDNAs, the following settings were used: 95 °C 2 min, 30 cycles of (95 °C 10 sec, 54 °C 15 sec, 72 °C 30 sec), 72 °C 5 min. For *act1* + cDNAs, 24 cycles with an elongation step of 15 sec were instead used. Samples were then loaded on BET-stained agarose gels, prior to imaging with a ChemiDoc MP Imaging System (BIORAD).

For RT-qPCR analyses, cDNAs were quantified with SYBR Green Master Mix (Roche, #04887352001) and a LightCycler LC480 apparatus (Roche).

Oligonucleotides used in PCR and qPCR reactions are listed in Supplementary Table 3.

## RNA-immunoprecipitation (RIP)

50 to 150 ODs of cells grown in EMM, EMM-LEU or EMM-N supplemented with L-leucine, NH$_4$Cl and 3MB-PP1 for 3 h after meiosis induction were typically used. Following addition of 1 mM PMSF (ThermoFisher scientific, #36978) to the cultures for 2 min, cells were harvested by centrifugation. Cell pellets were washed in 1X PBS, resuspended in 1.5 to 2.4 mL lysis buffer (6 mM Na$_2$HPO$_4$, 4 mM NaH$_2$PO$_4$, 150 mM NaC$_2$H$_3$O$_2$, 5 mM MgC$_2$H$_3$O$_2$, 0.25% NP-40, 2 mM EDTA, 1 mM EGTA, 5% glycerol, 1 mM PMSF, 10 U RNaseOUT Ribonuclease inhibitor (Invitrogen, #10777-019) per mL lysis buffer) and the mixture was slowly dropped in liquid nitrogen prior to cryolysis for 5 × 3 min at 10 Hz using a Ball Mill apparatus (Retsch, MM400). Extracts were then thawed and centrifuged twice (4000 g 5 min and 10000 g 10 min) before immunoprecipitations with 5 to 10 μL rabbit IgG-conjugated M270 epoxy Dynabeads (Invitrogen, #14311D) for 20 min at 4 °C. Following two washes in IPP150 (10 mM Tris pH8, 150 mM NaCl, 0.1% NP-40), total and immunoprecipitated RNAs were extracted with phenol:chloroform 5:1 pH4.7 (Sigma, #P1944) and ethanol-precipitated in the presence of 0.3 M sodium acetate and 20 μg glycogen. RNA samples were washed with 70% ethanol, resuspended in water and treated with DNase (Ambion, #AM1906) prior to Northern blotting analyses.

## RNAseH cleavage assay

2 μg of total, DNAse-treated RNAs precipitated with 0.3 M sodium acetate were heat-denatured at 94 °C for 3 min in the presence or absence of 2 μL RNA/DNA chimeric oligonucleotide at 1 μM (Supplementary Table 2). Samples were then incubated at 55 °C for 10 min prior to addition of 2U RNaseH (New England Biolabs, #M0297S) and further incubation at 55 °C for 5 min. Reactions were stopped in the presence of 100% ethanol, 0.3 M sodium acetate and 20 μg glycogen. After 1 h incubation at -80 °C, samples were centrifuged 30 min at 20,000 g and RNA pellets were washed with 70% ethanol and resuspended in 16 μL water.

## Northern blotting

1 to 4 μg of total RNAs (or 4-5 μL of total and immunoprecipitated RNAs from RIP experiments) were denatured at 65 °C for 5 min in the presence of 2X RNA loading dye (ThermoFisher Scientific, #R0641) and separated on 0.8% or 1.2% agarose gels in 1X TBE buffer. RNAs were transferred on a nitrocellulose membrane (Amersham, #RPN203B) by capillarity in 10X SSC buffer (1.5 M NaCl, 0.15 M Na Citrate – pH 7) for 6 h. RNAs were then crosslinked to the membrane by UV irradiation (2 x 1200 J, energy mode) using a dedicated apparatus. The membrane was next pre-hybridized for 30 min in hybridization buffer (DIG Easy Hyb Granules, Merck, #11796895001) at 65 °C. Heat-denatured, DIG-labeled RNA probe was subsequently added to the buffer prior to overnight hybridization at 65 °C. The membrane was then washed

twice in 2X SSC, 0.1% SDS for 10 min and twice in 0.1X SSC, 0.1% SDS for 15 min, at 65 °C, prior to incubation in washing buffer (0.1 M $C_4H_4O_4$, 0.15 M NaCl, 0.3% Tween-20 – pH7.5) for 4 min and in blocking solution (DIG Northern starter kit, Roche, #12039672910, diluted 1:10 in 0.1 M maleic acid, 0.15 M NaCl solution – pH 7.5) for 30 min at room temperature. 5 µL anti-DIG antibody (Anti-digoxigenin-AP fab fragments, Roche, #11093274910; RRID:AB 2734716) was next added to the blocking solution, and incubation lasts for 1.5 to 2 h. After 2 x 15 min washes in 0.1 M $C_4H_4O_4$, 0.15 M NaCl, 0.3% Tween-20 – pH7.5, the membrane was incubated with detection buffer (0.1 M Tris-HCl, 0.1 M NaCl – pH 9.5) for 5 min. Revelation was performed with a dedicated reagent (CDP-Star, Merck, #12041677001) and a ChemiDoc MP Imaging System (BIORAD).

DIG-labeled RNA probes were synthesized by in vitro transcription using appropriate DNA templates, 10X DIG RNA labeling mix (Merck, 11277073910), and T7 RNA polymerase (ThermoFisher Scientific, #EP0111). Samples were incubated for 2 h at 37 °C before the addition of DNAse (Roche, # 04716728001) for 15 min and 0.2 M EDTA to stop reactions.

Oligonucleotides used to generate the DNA templates are listed in Supplementary Table 3.

Uncropped scans are provided in the Source Data file.

### 5' and 3' RACE analyses
The 5'/3' RACE Kit, 2nd Generation (Roche, #3353621001), was used with the Transcriptor Reverse Transriptase Kit (Roche #03531317001) to determine the 5' and 3' ends of *snR107* as well as the 3' end of *mamRNA* upon defective splicing (5'&3'SSmut), following manufacturer's instructions. 5 µg of total RNA was polyadenylated using *E. coli* poly(A) polymerase (New England Biolabs, #M0276S) prior to 3' RACE.

Oligonucleotides used to amplify and clone RACE products are listed in Supplementary Table 3.

### Sequence analyses
Multiple sequence alignment of *snR107, snR190, SNORD12/B/C* was performed using CLUSTALW from GenomeNet using default settings. Nucleotides immediately upstream of the C box and downstream of the D box were excluded for the analysis.

Full-length *snR107* secondary structure prediction was generated using the UNAFold Web Server / RNA Folding Form[68] using the following constraints to maintain the ASE1 and ASE2 single-stranded: P 19 0 14 and P 83 0 11. Other default settings were applied.

### RNA-sequencing
Total RNA quality was assessed on an Agilent Bioanalyzer 2100, using RNA 6000 pico kit (Agilent, #5067-1513). 1 µg of total RNA was treated with Baseline-ZERO DNase (Biosearch technologies, #DB0715K) and subjected to ribosomal RNA depletion using the TruSeq Stranded Total RNA library prep Gold kit (Illumina, #20020598) according to the manufacturer's recommendations. Directional RNA-Seq Libraries were constructed using the TruSeq Stranded Total RNA library prep Gold kit (Illumina, #20020598), following the manufacturer's instructions. Libraries were pooled and sequenced on a Paired-End 51-35 bp run, on an Illumina NextSeq550 instrument. Demultiplexing was performed with bcl2fastq2 v2.18.12. Adapters were trimmed with Cutadapt v1.15, and only reads longer than 10pb were kept for further analysis.

Poly(A) + RNA-sequencing was performed following Plant and Animal Eukaryotic mRNA (WOBI) projects (Novogene, UK). After RNA sample quality control, about 200 ng of total RNA was used for poly(A) + enrichment with poly-T oligo-attached magnetic beads. mRNA libraries were generated using the Novogene NGS RNA Library Prep Set (PT042) and sequenced to a minimal depth of 30 million reads per sample, using a 150 bp paired-end sequencing strategy (NovaSeq 6000 S4 Reagent kit) on an Illumina NovaSeq 6000 instrument (Novogene).

Demultiplexing was performed with bcl2fastq v2.20 and adapters were trimmed with fastp v0.23.1.

Reads were aligned to the *S. pombe* genome using the STAR v2.6.1 d aligner[69]. Reads were quantified using featureCounts from subread v2.0.6, and RPKM was calculated using edgeR library from R. Heatmaps and box plots were generated using basic function in R.

### direct RNA-sequencing (dRNA-seq)
9 µg of total RNA was subjected to ribosomal RNA depletion using the TruSeq Stranded Total RNA library prep Gold kit (Illumina, #20020598). Following purification, 350 ng RNA was polyadenylated for 1.5 min at 37 °C with *E. coli* poly(A) polymerase (New England Biolabs, #M0276S). Libraries were prepared using the Direct RNA Sequencing kit (Oxford Nanopore Technologies, #SQK-RNA004) according to the manufacturer's recommendations and sequenced on PromethION RNA flowcells on the P2 solo instrument. Data were base-called on super accuracy mode using dorado 7.2.13-1. The sequences were mapped on the *S. pombe* genome with minimap2 with -x splice parameter.

### RiboMethSeq
About 150 ng of total RNA was subjected to random fragmentation by alkaline hydrolysis in 50 mM sodium-bicarbonate buffer (pH 9.2) at 96 °C for 16 min. The reaction was stopped by ethanol precipitation in the presence of 3 M sodium acetate (pH 5.2) and gly-coblue. After centrifugation, RNA pellets were washed with 80% ethanol and resuspended in nuclease-free water. RNA fragments were end-repaired as previously described[45] and purified using RNeasy MinElute Cleanup kit (QIAGEN, #74204) according to the manufacturer's recommendations, except that 675 µL 96% ethanol were used for RNA binding. Elution was performed in 19 µL nuclease-free water. RNA fragments were converted to library using the NEBNext® Small RNA Library Prep Set for Illumina® (New England Biolabs, #E7330S) following the manufacturer's recommendations. DNA library was quantified using a fluorometer (Qubit 3.0 fluorometer, Invitrogen) and qualified using a High Sensitivity DNA chip on Agilent Bioanalyzer 2100. Libraries were multiplexed and subjected to high-throughput sequencing on an Illumina Next-Seq2000 instrument with a 50 bp single-end read mode.

Raw sequencing reads were subjected to trimming using Trimmomatic[70] v0.39 with the following parameters: MINLEN:08, STRINGENCY:7, AVGQUAL:30. Trimmed reads were further processed to keep only short reads with fragmentation-defined 3'-extremities. Selected trimmed reads were aligned to the *S. pombe* rRNA reference sequence (NR_151396.1, NR_151436.1, NR_151429.1, NR_151433.1) using bowtie2[71] v2.4.4. Reads' extremities (5'- and 3'-) were counted for each RNA position in the reference and cumulative 5'/3'-protection profile was calculated as previously described[45,72] in R/R-studio environment. Specific RiboMethSeq scores (mean, A, B and C = MethScore[44,72]) were calculated in +/- 2 nt interval for all RNA positions and extracted for positions of known Nm residues in target RNA. If not available, potential Nm positions in RNA sequence were predicted using a medium stringency combination of score mean >0.95 and ScoreA > 0.5, this combination generally giving the best balance between sensitivity and selectivity in de novo Nm detection[73].

Note that, in data presented in Fig. 3a and Supplementary Fig. 3b, negative values were not shown for clarity and, due to the +/- 2 nt calculation interval, variations of MethScores at $G^{2483}$ neighboring positions (*i.e.* $G^{2481}$, $G^{2482}$, $G^{2484}$ and $A^{2485}$) were observed between wild type and *snR107* mutant cells.

### Total protein extraction and analyses
Total proteins were extracted from 4 to 5 mL of yeast cells grown in exponential phase as previously described[36,37], except that pellets were

resuspended in 40 to 50 μL SDS sample buffer (150 mM Tris pH 8.5, 4% SDS, 5% glycerol, bromophenol blue, 100 mM DTT) prior to heat denaturation at 95 °C for 10 min. Samples were analyzed by standard immunoblotting procedures using 1:3000 peroxydase-conjugated antiperoxydase (PAP, to detect protein-A-tagged proteins) (Sigma, #P1291, RRID:AB_1079562), 1:3000 anti-FLAG monoclonal antibody (Sigma, #F3165, RRID:AB_259529), 1:3000 anti-CDC2 monoclonal antibody (Abcam, #ab5467, RRID:AB_2074778) and 1:5000 goat anti-mouse IgG-HRP (Santa Cruz Biotechnology, #sc-2005, RRID:AB_631736). Detection was performed with SuperSignal West Pico Chemiluminescent Substrate (ThermoFisher Scientific, #34080), ECL Select reagent (GE Healthcare, #RPN2235) and a ChemiDoc MP Imaging System (BIORAD).

Uncropped scans are provided in the Source Data file.

## Co-immunoprecipitation

40 to 50 ODs of cells grown in EMM or EMM-N supplemented with L-leucine, NH$_4$Cl and 3MB-PP1 for 3 h after meiosis induction were used. Following addition of 1 mM PMSF (ThermoFisher scientific, #36978) to the cultures for 2 min, cells were harvested by centrifugation. Cell pellets were washed in 1X PBS, resuspended in 1.2 to 1.5 mL lysis buffer (6 mM Na$_2$HPO$_4$, 4 mM NaH$_2$PO$_4$, 150 mM NaC$_2$H$_3$O$_2$, 5 mM MgC$_2$H$_3$O$_2$, 0.25% NP-40, 2 mM EDTA, 1 mM EGTA, 5% glycerol, 1 mM PMSF) and the mixture was slowly dropped in liquid nitrogen prior to cryolysis for 5 x 3 min at 10 Hz using a Ball Mill apparatus (Retsch, MM400). Extracts were thawed and centrifuged twice (4000 g 5 min and 10000 g 10 min) before immunoprecipitation with 5 μL rabbit IgG-conjugated M270 epoxy Dynabeads (Invitrogen, #14311D) for 20 min at 4 °C. Beads were then washed twice in IPP150 (10 mM Tris pH8, 150 mM NaCl, 0.1% NP-40) and total and immunoprecipitated fractions were heat-denaturated at 95 °C for 10 min in the presence of SDS sample buffer (150 mM Tris pH 8.5, 4% SDS, 5% glycerol, bromophenol blue, 4% ß-mercaptoethanol). Standard immunoblotting procedures were performed using 1:3000 peroxydase-conjugated antiperoxydase (PAP) (Sigma, #P1291), 1:1000 anti-GFP monoclonal antibody (Roche, #11814460001, RRID:AB_390913), 1:3000 anti-HA monoclonal antibody (12CA5) (Sigma, #11583816001, RRID:AB_514505) 1:3000, anti-CDC2 monoclonal antibody (Abcam, #ab5467) and 1:5000 goat anti-mouse IgG-HRP (Santa Cruz Biotechnology, #sc-2005). Detection was performed with SuperSignal West Pico Chemiluminescent Substrate (ThermoFisher Scientific, #34080), ECL Select reagent (GE Healthcare, #RPN2235) and a ChemiDoc MP Imaging System (BIORAD). Uncropped scans are provided in the Source Data file.

## Polysome profiling

200 ODs of cells were grown before the addition of 200 μL ethanol-solubilized cycloheximide (Sigma, #C1988-5G) at 100 mg/mL for 5 min at 30 °C and 15 min in ice. Following centrifugation, cell pellets were washed with 10 mL polysome buffer (10 mM Tris-HCl pH7.5, 100 mM NaCl, 30 mM MgCl$_2$, 1 μL cycloheximide at 100 mg/mL) and resuspended in 900 μL of polysome buffer. The mixture was slowly dropped in liquid nitrogen prior to cryolysis for 3 x 3 min at 8 Hz using a Ball Mill apparatus (Retsch, MM400). Extracts were thawed and centrifuged twice (4000 g 5 min and 10,000 g 10 min) at 4 °C and absorbance units at 260 nm (RNA concentration) were measured using a DeNovix DS-11 device. The equivalent of 16 absorbance units was dropped at the surface of a 11 mL sucrose gradient prepared in Beckman tubes (#331372), composed of 30% sucrose, 50 mM Tris-acetate pH 7.6, 50 mM NH$_4$Cl, 12 mM MgCl$_2$, 1 mM DTT and frozen/thawed 3 times before use. Sample-containing gradients were then centrifuged for 3 h at 36,000 rpm, 4 °C and polysomes were analyzed using a Teledyne Isco device equipped with a Tris peristaltic pump (speed 30, scale 1) and a UA-6 UV/VIS Detector with type 11 optical unit.

## Microscopy analyses

To determine the percentages of cells with 1, 2 or 3 to 4 nuclei upon induction of synchronized meiosis, 1 OD of cells grown in EMM-N supplemented with L-leucine, NH$_4$Cl and 3MB-PP1 was harvested every h for up to 9 h and fixed with 100% ethanol prior to DNA staining with Hoescht 33342 (ThermoFisher Scientific, #62249). Cells were imaged at room temperature with a motorized Olympus BX63 upright fluorescence microscope equipped with a 60X and 100X oil immersion objective (Olympus), a digital camera C11440 (ORCA-Flash4.0 LT PLUS; Hamamatsu) and the CellSens software. DNA was visualized using a DAPI (CHROMA 49000 - ET- DAPI) filter set and images were processed in Fiji (NIH).

To assess sporulation efficiency, homothallic cells plated on ME plates for 3 days at 30 °C were imaged at room temperature in bright field using the same microscope device.

## Single molecule Fluorescence In Situ Hybridization (smFISH)

smFISH probes used in this study are listed in Supplementary Table 4. Quasar 570 or 670-labeled probe sets were designed manually or using Stellaris Probe Designer tool, and synthesized by Biosearch Technologies. smFISH was performed according to the manufacturer's protocol (Biosearch Technologies) with previously described modifications[37]. Cells were imaged using a Leica DM6000B microscope equipped with a 100X, NA 1.4 (HCX Plan-Apo) oil immersion objective, a piezo-electric motor (LVDT; Physik Instrument) mounted underneath the objective lens, and a CCD camera (CoolSNAP HQ; Photometrics). Maximum intensity-projection of optical Z-sections (0.2 μm, 25 planes) was performed with Metamorph (Molecular Devices).

Uncropped images are provided in the Source Data file.

## Statistics and reproducibility

n values were chosen in accordance with standard practices and correspond to the number of biological replicates (i.e. independent yeast cultures).

All experiments comprising RT-qPCR assays were repeated at least three times, involving n biological replicates as indicated in figure legends. qPCR measurements were statistically compared using two-tailed t-tests with the following p-value cut-offs for significance: $0.05 > * > 0.01; 0.01 > ** > 0.001; *** < 0.001$.

All representative images underlying Western and Northern blotting, live cell microscopy and smFISH analyses were obtained from experiments repeated at least two or three times involving independent biological replicates.

## Reporting summary

Further information on research design is available in the Nature Portfolio Reporting Summary linked to this article.

## Data availability

The RNA-seq and dRNA-seq data generated in this study have been deposited in NCBI's Gene Expression Omnibus database under GEO Series accession codes GSE276242, GSE276243 and GSE276244. The RiboMethSeq data generated in this study have been deposited in the European Nucleotide Archive database under accession code PRJEB79915. Source data are provided with this paper.

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

## Acknowledgements

We thank José Ayté, the Yeast Genetic Resource Center (YGRC) from National BioResource Project (Japan) and Addgene for providing yeast strains and plasmids. We are grateful to Isabelle Hatin for technical assistance in polysome profiling. We thank Nathalie Bonnefoy, Maria Costa, Olivier Namy and Carlo Yague-Sanz for fruitful discussions. We acknowledge the sequencing and bioinformatics expertise of the I2BC High-throughput sequencing facility, supported by France Génomique (funded by the French National Program "Investissement d'Avenir" ANR-10-INBS-09). EL was supported by a PhD fellowship from the French Ministry of Higher Education and Research (Université Paris-Saclay). This work was supported by the Centre National pour la Recherche Scientifique, the Fondation ARC (ARCPJA2021060003978 to MR) and the Agence Nationale de la Recherche (ANR-18-CE45-0005 to J.A; ANR-21-CE12-0029 to BP and MR).

## Author contributions

E.L. and D.C. conceptualized and designed the work, acquired, analyzed and interpreted data, and revised the manuscript. S.P., C.G., A.M., V.M., Y.J., E.v.D., D.N., J.A., Y.M., and B.P. acquired, analyzed and interpreted data and revised the manuscript. M.R. conceptualized and designed the work, acquired, analyzed and interpreted data, drafted and revised the manuscript and supervised the work.

## Competing interests

The authors declare no competing interests.
