## [Transparent Peer Review file · Nature Communications]

A bifunctional snoRNA with separable activities in guiding rRNA 2'-O-methylation and scaffolding gametogenesis effectors

Corresponding Author: Dr Mathieu Rougemaille

Version 0:

Reviewer comments:

Reviewer #1

(Remarks to the Author)

The manuscript by Leroy and colleagues describes a novel snoRNA snR107 in fission yeast and analyses its dual functions in ribosome genesis (modification of rRNA) and gametogenesis (via regulation of Mmi1 and Mei2 protein levels). This is very interesting and timely work, showing that snoRNA may have multiple functions different from canonical methylation and pseudouridylation.

While the first part of the manuscript describing the localization, processing and function of snR107 in rRNA modification is very clear and, in my opinion, does not require any corrections, the further sections raise some questions.

1. In particular, could the authors clarify the last observation regarding snR107-independent Fib1 interaction with Mei2 and Mmi1? Does this mean that other snoRNA also interact with Mei2 and Mmi1 and can regulate their levels?
2. Is there any competition in binding between Mei2/Mmi1 and Fib1 to snoRNA? Are snoRNA functional when bound by both or one of these proteins? If there is competition, is snR107 maturated/stable when not bound by Fib1?
3. I am also wondering if the lack of rRNA modification in G2483 (as all snR107 mutants seem to affect rRNA methylation) and so possibly somewhat defective ribosomes, drive changes in the expression of meiotic genes?

Overall, the manuscript is of high standard, and I recommend it for publishing in Nature Communications if the above points are addressed either experimentally or sufficiently explained and reasoned in the text.

Pawel Grzechnik

Reviewer #2

(Remarks to the Author)

Manuscript NCOMMS-24-63684 titled "A bifunctional snoRNA with separable activities in guiding rRNA 2'-O-methylation and scaffolding gametogenesis effectors" reports the identification and functional characterization in *S. pombe* of a novel snoRNA involved in ribosome synthesis and meiosis regulation. Using a combination of RNA-Seq, RT-qPCR, northern blotting, ONT and 5' and 3' RACE experiments, the authors convincingly show that mamRNA contains an intron hosting a previously unannotated box C/D snoRNA that they named snR107. This snoRNA is homologous to snR190 in *S. cerevisiae* and SNORD12/12B/12C in human cells. Besides the canonical C/D and C'/D' sequence elements, this snoRNA contains two antisense elements (ASE1 and ASE2) complementary to two regions of the 25S ribosomal RNA. The authors show that snR107 associates with box C/D snoRNP core proteins and guides ribose methylation of guanosine G2483 in the 25S rRNA via the ASE1 elements. Besides this, snR107 is required for efficient maturation of the 35S rRNA precursor and production of the 60S subunit. This function requires the K-loop motif and ASE2 element but is independent of the ASE1. Concerning meiosis, Mmi1 in association with mamRNA are required for Mei2 downregulation during vegetative growth to

avoid untimely expression of meiotic genes. Deletion of snR107, or mutation of its K-loop and ASE2 elements, affects Mei2 inhibition indicating that snR107 is the key determinant of Mei2 downregulation in *mamRNA*. Upon inhibition of the ubiquitin ligase Mot2 that stabilizes Mei2, snR107 via its K-loop and ASE2 elements prevents Mmi1 downregulation and the stabilization of meiotic transcripts during vegetative growth. Upon induction of meiosis, accumulation of Mei2 and meiRNA sequester Mmi1 to stabilize meiotic transcripts. In this context, the absence of snR107 changes the expression profile of meiotic genes, which are induced earlier, stronger and over a longer time window compared to control. The authors further show that Mmi1 and Mei2 interact with snR107 and Fib1 and that deletion of the K-loop sequence abrogates these interactions.

The demonstration that a small nucleolar RNA is potentially directly involved in the regulation of gametogenesis in *S. pombe* adds to the growing list of snoRNAs with regulatory functions outside ribosome synthesis, which is both timely and important to the broad fields of non-coding RNAs and meiosis regulation. The manuscript is very well written, reads well and is easy to grasp. The quality of the figures is very good and most of the conclusions are supported by the data presented. However, my main concern before accepting the manuscript for publication is that some of the meiotic phenotypes observed in the absence of snR107 might result from indirect ribosome synthesis and/or translation defects. As described below, the authors should provide additional experiments and controls to strengthen the conclusion that snR107 is directly and specifically involved in meiosis regulation.

Major points:

Fig. 3d and Supplementary Fig. 3c: the authors show using northern blotting and polysome profiling experiments that snR107 deletion affects ribosome synthesis and production of the 60S ribosomal subunit, resulting in translation defects. One important aspect that is not addressed in the manuscript is what are consequences of these defects on *S. pombe* growth rate and could these defects be responsible for meiosis deregulation observed later in the study (see below).

Fig. 4: the authors show that deletion of snR107 or any snR107 mutation affecting ribosome synthesis, results in Mei2 accumulation. It is not clear from the data whether this effect arises due to the perturbation of a direct function of snR107 in the regulation of Mei2 expression or due to indirect translation defects with potential pleiotropic consequences on gene expression, including a potential deregulation of the CCR4-NOT complex. The authors should provide controls showing that any other mutation in snoRNAs or ribosome biogenesis factors inducing similar defects as those observed upon snR107 deletion does not induce Mei2 accumulation. Furthermore, how about expression of meiRNA? Is it also expressed in these conditions?

Fig. 5: in cells undergoing meiosis, snR107 deletion alters the expression profile of meiotic genes. It is not clear from the data whether these changes are due to a direct regulatory function of snR107 during meiosis or to indirect effects. Among the possible hypotheses, as snR107 deletion increases the levels of Mei2 during vegetative growth, could the changes in expression of the meiotic genes be due to the higher level of Mei2 at the onset of meiosis? Another possibility could be that the changes in expression result from the reduced translational capacity of the cells lacking snR107. Although in both cases the intuitive expected effects would yield opposite data, the authors should experimentally rule out these possibilities.

Fig. 6b,c: Mmi1 and Mei2 interact with snR107 snoRNA but how specific is this interaction? The authors should assess whether other snoRNAs are present or not in the purified material. This aspect is important as the authors show in Supplementary Fig. 6 that Mmi1 and Mei2 interact with Fib1 independently of snR107.

Fig. 6e: mutation of the K-loop region of snR107 affects the association of snR107 with Mei2, Mmi1 and Fib1. They conclude in the following paragraph that Mmi1 and Mei2 are engaged in both protein-protein and protein-RNA interactions with the snoRNP. This conclusion is potentially risky as deletion of the K-loop sequence is expected to affect binding of the Snu13 protein to the C'/D' motif, which in turn prevents the stable association of Fib1 (as observed in the figure). This is also consistent with Fig. 3c showing that deletion of the K-loop sequence of snR107 abrogates methylation of G2483 guided by the C'/D' motif and the ASE2. Deletion of the K-loop is expected to induce drastic changes in the protein composition of the snoRNP, which may be sufficient to explain the loss of interaction with Mmi1 and Mei2.

Supplementary Fig. 6b,c: the interaction between Fib1 and Mmi1 and Mei2 does not depend on snR107. Two questions need to be clarified in that case: Do Mmi1 and Mei2 interact specifically with snR107 and if so, what determines the specificity of the interaction as the K-loop motif associated to the C' and D' boxes are common to many snoRNAs.

Minor points:

Supplementary Fig. 2b: the K-turn motif of box C/D snoRNAs is a precise RNA structure including a short stem and an asymmetric bulge containing two non-canonical G-A base pairs involving nucleotides of the C/D or C'/D' boxes. The K-turn motif is not visible on the Mfold secondary structure presented, because Mfold does not integrate the specificities of box C/D snoRNAs. The authors should better identify the K-turn motif of snR107 and determine if it deviates or not from the consensus structure as it is important for different conclusions of the manuscript.

Supplementary Fig. 2e: snR107 levels increase in the absence of Rrp6 and Cid14 but no precursor species are detected contrary to the absence of Dhp1. On this basis, and as the debranched lariat intron contains both 5' and 3' extensions, it is not clear whether Rrp6/Cid14 are indeed required for snR107 processing.

Reviewer #3

(Remarks to the Author)

The authors present evidence that a smaller, intron encoded fragment of the non-coding mamRNA not only plays a pivotal role in its previously characterised function controlling stability of meiotic mRNAs, but also functions as a Box C/D snoRNA in the 2'O methylation of the 25S rRNA in fission yeast. This newly characterized snoRNA (snR107) consequently expands the repertoire of functions linked to Box C/D snoRNAs.

The manuscript is very well written, and the data are compelling. It is very interesting that a snoRNA would have a dual functionality as has been proposed in the paper. I am enthusiastic about this paper.

Comments:

- The paper as presented relies mainly on deletion/loss of function data; i.e. the loss of 2'O-Me function or effects on Mei2/mRNA stability in the context of various mamRNA deletions or point mutants relating to snR170 splicing. Since the paper is proposing that the 2'-O-Me and meiotic mRNA decay associated functions are controlled by a subset of the previously characterized, longer mamRNA, support for the proposed mechanism would be augmented by complete deletion of mamRNA followed by rescue of all functions by a minimal construct. Ideally this would be plasmid encoded, containing only snR170, and inducible to demonstrate rescue of functions in a background where associated functions were previously missing pre-induction. Alternatively the authors should at least generate a strain that contains only snR170, which is currently lacking, and confirm this is sufficient for associated functions. This would augment the claim in the manuscript that snR170 is at the heart of the proposed function as the authors claim (title and Figure 7), as opposed to a "necessary but not necessarily sufficient" argument which is more the current narrative.
- The paper would also be augmented with some work adding clarity as to why a snoRNA might function in an Mmi1/Ccr4-Not containing complex to control meiotic mRNA gene expression. The most straightforward explanation is that the fibrillarin associated 2'O methylation machinery also has an unanticipated role in Mmi1/Mei2 meiotic mRNA gene regulation. Is Mmi1/Mei2 associated control of meiotic mRNA decay impaired in a fib1 mutant/knockout? Or possibly in other mutants of the box C/D methylation guide snoRNP complex?
- The function(if any) of the 5' exon detected by probe A (Fig 1C) is unclear. Is it meaningful that it accumulates visibly by Northern (probe A, lane 1, wt, two bands?). The focus on localization of this fragment in 1D is also unclear; based on the narrative of the manuscript it seems it would have been more informative to identify the location of snR170.
- The IGV reads maps in Figure 2A need a better explanation for the significance between the top box and bottom box. There is no explanation for the meaning of the blue bars.
- The difference between lanes 1 and 2 in Figure 4B/C/D is not apparent. Is lane 1 the untagged strain?

Version 1:

Reviewer comments:

Reviewer #1

(Remarks to the Author)

The authors addressed my questions. I recommend the manuscript for publishing.

Pawel Grzechnik

Reviewer #2

(Remarks to the Author)

Revised manuscript NCOMMS-24-63684A titled "A bifunctional snoRNA with separable activities in guiding rRNA 2'-O-methylation and scaffolding gametogenesis effectors" has been substantially improved and the authors have addressed the most important concerns of the three reviewers. Concerning my own comments (reviewer #2), the authors now show in the revised manuscript that Mei2 accumulation in snR107 mutants is not due to indirect ribosome synthesis or translation defects as inactivation of other snoRNAs or the ribosome biogenesis factor *cgr1* does not induce Mei2 accumulation. Following meiosis induction, expression of meiotic transcripts in *cgr1*Δ cells is comparable to that observed in wild type cells, indicating that defects in ribosome assembly and/or activity do not indirectly change meiotic gene expression. The authors further show in the revised manuscript that Mmi1 and Mei2 interact with other snoRNAs besides snR107, in agreement with previous reports. This result rationalizes the snR107-independent interaction of Mmi1 and Mei2 with Fib1. In my opinion, the manuscript is now suitable for publication in Nature Communications journal.

Reviewer #3

(Remarks to the Author)

My concerns have been fully addressed in the revised manuscript through the response to reviewers as well as significant new work and revisions to the manuscript. I support its publication.

REVIEWER COMMENTS

We are grateful to the reviewers for their positive assessment of our work and their valuable comments and suggestions. We have revised the manuscript and performed additional experiments/controls, which further support the main conclusions of our study. Our point-by-point responses appear in blue below.

Reviewer #1 (Remarks to the Author):

The manuscript by Leroy and colleagues describes a novel snoRNA snR107 in fission yeast and analyses its dual functions in ribosome genesis (modification of rRNA) and gametogenesis (via regulation of Mmi1 and Mei2 protein levels). This is very interesting and timely work, showing that snoRNA may have multiple functions different from canonical methylation and pseudouridylation.

While the first part of the manuscript describing the localization, processing and function of snR107 in rRNA modification is very clear and, in my opinion, does not require any corrections, the further sections raise some questions.

1. In particular, could the authors clarify the last observation regarding snR107-independent Fib1 interaction with Mei2 and Mmi1? Does this mean that other snoRNA also interact with Mei2 and Mmi1 and can regulate their levels?

We thank the reviewer for pointing this out. We indeed observed that both Mmi1 and Mei2 still associate with Fib1 when their binding to *snR107* is decreased or abolished (i.e. in *snR107^{kloopmut}* cells, Supplementary Fig. 6c,d), suggesting that they associate with additional snoRNAs. Supporting this, we now provide evidence that both proteins interact through their RNA-binding domains with the mature form of U14 (new **Supplementary Fig. 6e,f**). This is also consistent with previous genomic approaches from our and other labs showing that Mmi1 and Mei2 bind to snoRNA species (PMID 26670050; 30257894; 33536434). Although the biological relevance of such interactions still remains unknown, it is possible that other snoRNAs, including U14, contribute to the regulation of Mmi1 and Mei2 activities. Nonetheless, our findings clearly indicate that *snR107* is necessary and sufficient for the control of Mei2 abundance and Mmi1-targeted meiotic mRNA levels (see also response to reviewer #3). Whether other snoRNAs may also regulate these processes in a non-redundant manner is an interesting scenario for future work.

2. Is there any competition in binding between Mei2/Mmi1 and Fib1 to snoRNA? Are snoRNA functional when bound by both or one of these proteins? If there is competition, is snR107 maturated/stable when not bound by Fib1?

We assessed the association of Fib1 with *snR107* in the absence of Mmi1 and found no further binding compared to wild type cells, indicating that Mmi1 does not antagonize Fib1 interaction with the snoRNA (new **Supplementary Fig. 6a**). Although we could not perform the reverse experiment due to the essential role of Fib1 for cell viability, these results suggest that the proteins do not compete for interaction with *snR107*.

Our data indicate that mutation of the *snR107* k-loop decreases/abolishes the binding of Mmi1, Mei2 and Fib1 and disrupts its functions independently of expression or localization defects. However, it remains unclear at this stage whether Mmi1/Mei2 modulate Gm²⁴⁸³ deposition (i.e. Fib1 activity) on one side, and conversely, whether Fib1 or its partners are required for the

regulations of Mmi1 and Mei2 on the other side (see also response to reviewer #3). We will surely consider studying such possible cross-talks in the future.

3. I am also wondering if the lack of rRNA modification in G2483 (as all snR107 mutants seem to affect rRNA methylation) and so possibly somewhat defective ribosomes, drive changes in the expression of meiotic genes?

We have shown that the lack of Gm²⁴⁸³ *per se* (i.e. in snR107_{ASE1mut} cells) does not impact ribosome biogenesis nor meiotic gene expression profiles, excluding the possibility that altered expression of Mmi1 mRNA targets in other snR107 mutants is causally linked to this rRNA modification (Fig. 3d, Fig. 5e). However, it remained possible that the changes in meiotic gene expression observed in snR107_{kloopmut} and snR107_{ASE2mut} cells result from defective ribosome biogenesis. To address this point (see also response to reviewer #2), we assessed the functional consequences of inactivating Cgr1, a factor involved in the biogenesis of pre-60S ribosomal particles. As shown in the revised version of the manuscript, cells lacking Cgr1 exhibited strong defects in ribosome biogenesis (new **Supplementary Fig. 4e**) but the expression kinetics of Mmi1-targeted meiotic genes upon synchronized meiosis were generally comparable to those observed in *wt* cells (new **Supplementary Fig. 5b**). Although the induction of *mcp5+* and *ssm4+* mRNAs was slightly longer in *cgr1Δ* as compared to *wt*, these experiments clearly indicate that defective ribosome biogenesis is not the primary cause of increased meiotic gene expression in *snR107* mutants accumulating Mei2.

Overall, the manuscript is of high standard, and I recommend it for publishing in Nature Communications if the above points are addressed either experimentally or sufficiently explained and reasoned in the text.

We thank the reviewer for the positive comments and hope that we satisfied the points raised.

Reviewer #2 (Remarks to the Author):

Manuscript NCOMMS-24-63684 titled "A bifunctional snoRNA with separable activities in guiding rRNA 2'-O-methylation and scaffolding gametogenesis effectors" reports the identification and functional characterization in *S. pombe* of a novel snoRNA involved in ribosome synthesis and meiosis regulation. Using a combination of RNA-Seq, RT-qPCR, northern blotting, ONT and 5' and 3' RACE experiments, the authors convincingly show that mamRNA contains an intron hosting a previously unannotated box C/D snoRNA that they named snR107. This snoRNA is homologous to snR190 in *S. cerevisiae* and SNORD12/12B/12C in human cells. Besides the canonical C/D and C'/D' sequence elements, this snoRNA contains two antisense elements (ASE1 and ASE2) complementary to two regions of the 25S ribosomal RNA. The authors show that snR107 associates with box C/D snoRNP core proteins and guides ribose methylation of guanosine G2483 in the 25S rRNA via the ASE1 elements. Besides this, snR107 is required for efficient maturation of the 35S rRNA precursor and production of the 60S subunit. This function requires the K-loop motif and ASE2 element but is independent of the ASE1. Concerning meiosis, Mmi1 in association with mamRNA are required for Mei2 downregulation during vegetative growth to avoid untimely expression of meiotic genes. Deletion of snR107, or mutation of its K-loop and ASE2 elements, affects Mei2 inhibition indicating that snR107 is the key determinant of Mei2 downregulation in mamRNA. Upon inhibition of the ubiquitin ligase Mot2 that stabilizes Mei2, snR107 via its K-loop and ASE2 elements prevents Mmi1 downregulation and the stabilization of meiotic transcripts during vegetative growth. Upon induction of meiosis, accumulation of Mei2 and meiRNA sequester Mmi1 to stabilize meiotic transcripts. In this context, the absence of snR107 changes the expression profile of meiotic genes, which are induced earlier, stronger and over a longer time window compared to control. The authors further show that Mmi1 and Mei2 interact with snR107 and Fib1 and that deletion of the K-loop sequence abrogates these interactions.

The demonstration that a small nucleolar RNA is potentially directly involved in the regulation of gametogenesis in *S. pombe* adds to the growing list of snoRNAs with regulatory functions outside ribosome synthesis, which is both timely and important to the broad fields of non-coding RNAs and meiosis regulation. The manuscript is very well written, reads well and is easy to grasp. The quality of the figures is very good and most of the conclusions are supported by the data presented. However, my main concern before accepting the manuscript for publication is that some of the meiotic phenotypes observed in the absence of snR107 might result from indirect ribosome synthesis and/or translation defects. As described below, the authors should provide additional experiments and controls to strengthen the conclusion that snR107 is directly and specifically involved in meiosis regulation.

We appreciate the positive comments of the reviewer. As requested, we have performed additional experiments and controls to exclude that increased meiotic mRNA expression and Mei2 levels in *snR107* mutants are linked to defective ribosome biogenesis and/or translation.

Major points:

Fig. 3d and Supplementary Fig. 3c: the authors show using northern blotting and polysome profiling experiments that snR107 deletion affects ribosome synthesis and production of the 60S ribosomal subunit, resulting in translation defects. One important aspect that is not addressed in the manuscript is what are consequences of these defects on *S. pombe* growth rate and could these defects be responsible for meiosis deregulation observed later in the study (see below).

We assessed the growth of all *snR107* mutants used in our study, including *snR107* Δ , *snR107*_{ASE1mut}, *snR107*_{kloopmut}, *snR107*_{21nt Δ} and *snR107*_{ASE2mut}. As shown in new **Supplementary Fig. 3g**, the different strains grew similarly when compared to wild type cells, indicating that the defects in rRNA modification and/or ribosome biogenesis have only marginal consequences, if at all, on mitotic proliferation. Although subtle differences cannot be formally excluded, the delays in meiosis progression and the altered meiotic gene expression profiles are thus unlikely to be due to defective fitness of *snR107* mutants.

Fig. 4: the authors show that deletion of *snR107* or any *snR107* mutation affecting ribosome synthesis, results in *Mei2* accumulation. It is not clear from the data whether this effect arises due to the perturbation of a direct function of *snR107* in the regulation of *Mei2* expression or due to indirect translation defects with potential pleiotropic consequences on gene expression, including a potential deregulation of the CCR4-NOT complex. The authors should provide controls showing that any other mutation in snoRNAs or ribosome biogenesis factors inducing similar defects as those observed upon *snR107* deletion does not induce *Mei2* accumulation. Furthermore, how about expression of *meiRNA*? Is it also expressed in these conditions?

We thank the reviewer for pointing this out. We first generated strains lacking C/D-box snoRNAs that guide 2'-O-methylation of 25S rRNA residues located in domain V, alike *Gm*²⁴⁸³. We deleted *snR88*, *snoR69* and *snoR69b*, which target *Um*²³⁰⁵ and *Um*²⁴⁹⁸, *Am*³⁰⁴¹ and *Cm*³⁰⁴³, respectively. However, none of these deletions, alone or in combination, resulted in decreased levels of 60S/80S particles and the appearance of half-mers (please see attached figure below), contrary to what observed in *snR107* mutants. The absence of phenotypes in the different mutant strains therefore highlight an important and peculiar function for *snR107* in ribosome biogenesis.

Figure. Polysome profiles from total cellular extracts obtained on sucrose gradients in strains of the indicated genetic backgrounds. Absorbance at 254 nm was measured for up to 13 min of collection time and is expressed as arbitrary units (A.U).

Since we could not use the above-mentioned snoRNA KOs to address the reviewer's comment, we investigated the impact of *Cgr1*, a non-essential factor previously involved in pre-rRNA

processing and biogenesis of pre-60S ribosomal particles in budding yeast (PMID 11932453, 30291245). Consistent with this, *cgr1*Δ cells exhibited a strong reduction in the levels of monosomes (80S) and pronounced half-mers, to an extent that was even higher than that of the *snR107*Δ mutant (new **Supplementary Fig. 4e**). However, inactivation of Cgr1 did not result in the accumulation of Mei2, as opposed to *snr107* deletion (new **Supplementary Fig. 4f**). These data therefore indicate that the increased levels of Mei2 in relevant *snR107* mutants are not the mere consequence of indirect defects in ribosome biogenesis and/or activity.

As requested, we have further evaluated *meiRNA* expression in cells lacking Mot2 and/or mutated for *snR107*. Similar to meiotic mRNAs (e.g. *mei4+*, *mcp5+*, *ssm4+*), *meiRNA* levels increased in the *mot2*Δ mutant in a *snR107*-dependent manner (new **Fig. 4f**). Importantly, *snR107* role in *meiRNA* regulation exhibits the same genetic requirements as for Mei2/Mmi1 regulations, i.e. necessitating both k-loop and ASE2 elements (new **Fig. 4f**).

We have included and commented these different results in the revised version of the manuscript.

Fig. 5: in cells undergoing meiosis, *snR107* deletion alters the expression profile of meiotic genes. It is not clear from the data whether these changes are due to a direct regulatory function of *snR107* during meiosis or to indirect effects. Among the possible hypotheses, as *snR107* deletion increases the levels of Mei2 during vegetative growth, could the changes in expression of the meiotic genes be due to the higher level of Mei2 at the onset of meiosis? Another possibility could be that the changes in expression result from the reduced translational capacity of the cells lacking *snR107*. Although in both cases the intuitive expected effects would yield opposite data, the authors should experimentally rule out these possibilities.

We thank the reviewer for this comment. Our data indicate that deletion or specific mutations of *snR107* impact the timing and amplitude of Mmi1-targeted meiotic mRNAs upon synchronized meiosis. In line with the first proposed possibility, we have now clarified that *snr107* effect on meiotic genes likely involves changes in Mei2/Mmi1 levels and/or activities. To exclude the second possibility, i.e. that changes in meiotic gene expression are indirect and result from altered ribosome biogenesis and/or reduced translational efficiency, we determined meiotic transcript profiles in *cgr1*Δ cells following *pat1*-induced ectopic meiosis. As shown in new **Supplementary Fig. 5b**, both *mcp5+* and *ssm4+* mRNAs accumulated similarly to what observed in wild type cells, albeit slightly longer. These data thus indicate that defects in ribosome assembly and/or activity, as scored in *cgr1*Δ cells, do not generally cause an increase in meiotic gene expression during synchronized meiosis and further point to a specific, Mmi1 and Mei2-related regulatory function of *snR107* during this cell cycle. We have included these results in the revised version of the manuscript and amended the text accordingly.

Fig. 6b,c: Mmi1 and Mei2 interact with *snR107* snoRNA but how specific is this interaction? The authors should assess whether other snoRNAs are present or not in the purified material. This aspect is important as the authors show in Supplementary Fig. 6 that Mmi1 and Mei2 interact with Fib1 independently of *snR107*.

Our data indeed indicate that Mmi1 and Mei2 associate with Fib1 in a *snR107*-independent manner, suggesting that both proteins may also interact with other C/D-box snoRNAs. We assessed the presence of mature U14 in the purified material and indeed found that Mmi1 and Mei2 interact with it (new **Supplementary Fig. 6e,f**; see also response to reviewer #1). Further supporting this notion, previous genome-wide analyses revealed that both proteins associate

with several snoRNA species, including U14 (PMID 26670050; 30257894; 33536434), thereby rationalizing the *snR107*-independent interaction between Mmi1/Mei2 and Fib1. We have amended the text in this perspective.

Fig. 6e: mutation of the K-loop region of snR107 affects the association of snR107 with Mei2, Mmi1 and Fib1. They conclude in the following paragraph that Mmi1 and Mei2 are engaged in both protein-protein and protein-RNA interactions with the snoRNP. This conclusion is potentially risky as deletion of the K-loop sequence is expected to affect binding of the Snu13 protein to the C'/D' motif, which in turn prevents the stable association of Fib1 (as observed in the figure). This is also consistent with Fig. 3c showing that deletion of the K-loop sequence of snR107 abrogates methylation of G2483 guided by the C'/D' motif and the ASE2. Deletion of the K-loop is expected to induce drastic changes in the protein composition of the snoRNP, which may be sufficient to explain the loss of interaction with Mmi1 and Mei2.

We apologize for not being clear enough in the original version of the manuscript. The k-loop sequence lying between the C' and D' boxes was not deleted but substituted by its reverse complement (CGACUA to UAGUCG). Regardless, we agree with the reviewer that mutation of these six nucleotides in snR107_{kloopmut} cells may impair the binding of Snu13, possibly affecting in turn Fib1 association. However, the interaction of Nop58 is preserved in this context and neither *snR107* expression levels nor localization are affected, indicating that the snoRNP is still partially assembled. The decreased associations of Mmi1 and Mei2 in snR107_{kloopmut} cells may hence reflect a role for Fib1 (and possibly Snu13) in their recruitment rather than being due to a complete disruption of the RNP. We have clarified this aspect in the text.

Supplementary Fig. 6b,c: the interaction between Fib1 and Mmi1 and Mei2 does not depend on snR107. Two questions need to be clarified in that case: Do Mmi1 and Mei2 interact specifically with snR107 and if so, what determines the specificity of the interaction as the K-loop motif associated to the C' and D' boxes are common to many snoRNAs.

As mentioned above, Mmi1 and Mei2 associate with other snoRNAs, including U14 (new **Supplementary Fig. 6e,f**), consistent with previous genome-wide analyses (PMID 26670050; 30257894; 33536434). This is the likely reason why both proteins interact with Fib1 in a *snR107*-independent manner. We have mentioned this aspect in the revised version of the manuscript.

With respect to the specificity of the interaction, the k-loop motif and the surrounding degenerate C' and D' boxes may indeed adopt a common structure found in other snoRNAs. However, the k-loop sequence *per se* (i.e. UCGACTA), which is substituted in snR107_{kloopmut} cells, strongly varies between snoRNA species, both in nucleotide composition and length. The UCGACTA motif may hence provide some degree of specificity to the reported interactions, at the sequence and/or structural levels. Future work aimed at understanding the precise folding and organization of the *snR107*-containing RNP will illuminate the molecular bases of the interactions involving Mmi1 and Mei2.

Minor points:

Supplementary Fig. 2b: the K-turn motif of box C/D snoRNAs is a precise RNA structure including a short stem and an asymmetric bulge containing two non-canonical G-A base pairs involving nucleotides of the C/D or C'/D' boxes. The K-turn motif is not visible on the Mfold secondary structure presented, because Mfold does not integrate the specificities of box C/D snoRNAs. The authors should better identify the K-turn motif of snR107 and determine if it

deviates or not from the consensus structure as it is important for different conclusions of the manuscript.

We thank the reviewer for this observation. We agree that the Mfold structure prediction does not take into account non-canonical A-G base pairs. However, we think that hand manipulation of the *snR107* folding to accommodate potential A-G pairing is not appropriate. Only structural approaches may reveal the exact configuration of the k-turn motif. For this reason, we kept the original structure prediction while mentioning in the corresponding figure legend that A-G base pairs may contribute to the folding of both k-turn motifs (i.e. between C/D and C'/D' boxes).

Supplementary Fig. 2e: *snR107* levels increase in the absence of Rrp6 and Cid14 but no precursor species are detected contrary to the absence of Dhp1. On this basis, and as the debranched lariat intron contains both 5' and 3' extensions, it is not clear whether Rrp6/Cid14 are indeed required for *snR107* processing.

We apologize for not being clear enough on this aspect. We initially meant that Rrp6 and Cid14 contribute to *snR107* turn-over, not processing. We have rephrased the sentence to make clear that Dhp1 mediates *snR107* processing, while Rrp6 and Cid14 rather contribute to its decay.

Reviewer #3 (Remarks to the Author):

The authors present evidence that a smaller, intron encoded fragment of the non-coding *mamRNA* not only plays a pivotal role in its previously characterised function controlling stability of meiotic mRNAs, but also functions as a Box C/D snoRNA in the 2'O methylation of the 25S rRNA in fission yeast. This newly characterized snoRNA (*snR107*) consequently expands the repertoire of functions linked to Box C/D snoRNAs. The manuscript is very well written, and the data are compelling. It is very interesting that a snoRNA would have a dual functionality as has been proposed in the paper. I am enthusiastic about this paper.

We are grateful to the reviewer for the positive comments on our work.

Comments:

- The paper as presented relies mainly on deletion/loss of function data; i.e. the loss of 2'O-Me function or effects on *Mei2*/mRNA stability in the context of various *mamRNA* deletions or point mutants relating to *snR170* splicing. Since the paper is proposing that the 2'-O-Me and meiotic mRNA decay associated functions are controlled by a subset of the previously characterized, longer *mamRNA*, support for the proposed mechanism would be augmented by complete deletion of *mamRNA* followed by rescue of all functions by a minimal construct. Ideally this would be plasmid encoded, containing only *snR107*, and inducible to demonstrate rescue of functions in a background where associated functions were previously missing pre-induction. Alternatively the authors should at least generate a strain that contains only *snR107*, which is currently lacking, and confirm this is sufficient for associated functions. This would augment the claim in the manuscript that *snR107* is at the heart of the proposed function as the authors claim (title and Figure 7), as opposed to a "necessary but not necessarily sufficient" argument which is more the current narrative.

We thank the reviewer for this suggestion. We first attempted to only express *snR107* from a plasmid but failed to detect the mature form of the snoRNA, which remained trapped in a longer transcript (please see attached figure). This suggested that the intronic location of *snR107* is critical for its cellular accumulation. We therefore generated a plasmid expressing the sole *mamRNA* intron, which was sufficient to produce large amounts of mature *snR107* (new **Supplementary Fig. 4a**). Importantly, this plasmid-borne version of *mamRNA* intron (and hence mature *snR107*) restored low *Mei2* levels in otherwise *mamRNA*-deleted cells (i.e. *mamRNA*_{(11-737)Δ}), indicating that *snR107* is necessary and sufficient for the control of *Mei2* abundance (new **Fig. 4c**). We followed the same strategy to determine the importance of *snR107* in the regulation of meiotic

Figure. Northern blot showing *snR107* levels from total RNA samples, in the indicated genetic backgrounds. Ribosomal RNAs served as loading control.

mRNA expression upon inactivation of Mmi1 by high Mei2 levels (i.e. *mot2Δ* mitotic cells). As shown in **Supplementary Fig. 4g**, expression of plasmid-encoded *mamRNA* intron restored increased levels of Mmi1 RNA targets (i.e. *mei4+*, *ssm4+*, *mcp5+* mRNAs and *meiRNA*) in *mot2Δ* cells lacking *mamRNA* (i.e. *mamRNA*_{(11-737)Δ}), further supporting the notion that *snR107* is sufficient to modulate Mmi1 activity. We have added these results to the revised version of the manuscript, which together demonstrate that *snR107* is the critical regulatory effector within *mamRNA* involved in the Mmi1-Mei2 mutual control.

- The paper would also be augmented with some work adding clarity as to why a snoRNA might function in an Mmi1/Ccr4-Not containing complex to control meiotic mRNA gene expression. The most straightforward explanation is that the fibrillarin associated 2'O methylation machinery also has an unanticipated role in Mmi1/Mei2 meiotic mRNA gene regulation. Is Mmi1/Mei2 associated control of meiotic mRNA decay impaired in a *fib1* mutant/knockout? Or possibly in other mutants of the box C/D methylation guide snoRNP complex?

We thank the reviewer for pointing this out. We agree that our results may suggest an unexpected role for canonical snoRNP components in the control of Mmi1 and Mei2 activities. Unfortunately, all four subunits (i.e. Fib1, Snu13, Nop56 and Nop58) are essential for yeast viability, precluding the construction of deletion strains. To overcome this issue, we attempted to deplete Fib1 using the same auxin-inducible degron tag used to lower Dhp1 steady state levels. However, this approach was unsuccessful, as Fib1 still substantially accumulated from cells treated with various concentrations of auxin, possibly reflecting inefficient nucleolar targeting (please see attached figure below).

Figure. Western blot showing the levels of 3mAID-5xFLAG-tagged Dhp1 and Fib1 from cells treated with 1 mM (1x) or 4 mM (4x) indole acetic acid (IAA) for 2 hours. An anti-CDC2 antibody was used as loading control.

The possibility that Fib1 and/or its associated partners directly contribute to the regulation of Mmi1 and Mei2 is fascinating and we will surely dedicate more time and efforts in the future in this perspective.

- The function(if any) of the 5' exon detected by probe A (Fig 1C) is unclear. Is it meaningful that it accumulates visibly by Northern (probe A, lane 1, wt, two bands?). The focus on localization of this fragment in 1D is also unclear; based on the narrative of the manuscript it seems it would have been more informative to identify the location of snR170.

In our previous work reporting the identification of *mamRNA* (PMID 33536434), we showed that the lncRNA accumulates as two distinct isoforms that are resistant to Mmi1-dependent degradation, which might explain at least partially its detectable levels in wild type mitotic cells. Regardless of the precise mechanisms underlying its abundance, we also originally

unveiled the prominent localization of *mamRNA* 5' exon in one of the scattered Mmi1 nuclear foci in mitotic cells (PMID 33536434). The identification of a previously unannotated intron (this study, Fig 1a-c) therefore prompted us to re-evaluate this colocalization in cells defective for *mamRNA* splicing (i.e. 5'&3'SSmut cells), whereby the two lncRNA isoforms are trapped in a longer precursor transcript. We have clarified this in the revised version of the manuscript. As for the function of *mamRNA* 5' exon, we now provide evidence that it is not responsible for the regulations of Mmi1 and Mei2, which strictly depend on intron-encoded *snR107* (see above). Whether *mamRNA* 5' exon, a fraction of which is exported to the cytoplasm (PMID 33536434), may exert additional functions in mitotic and/or meiotic cells will require further investigation.

- The IGV reads maps in Figure 2A need a better explanation for the significance between the top box and bottom box. There is no explanation for the meaning of the blue bars.

We have now explained the differences between boxes (top: read coverage; bottom: sub sample of 200 reads from the locus and representative of the different read populations) and the meaning of blue bars (intronic sequences absent in corresponding reads) in the figure legend.

- The difference between lanes 1 and 2 in Figure 4B/C/D is not apparent. Is lane 1 the untagged strain?

Lane 1 indeed corresponds to the untagged strain. To increase clarity, we have added dashed lines between lanes 1 and 2 in the different panels.

REVIEWERS' COMMENTS

Reviewer #1 (Remarks to the Author):

The authors addressed my questions. I recommend the manuscript for publishing.

Pawel Grzechnik

We thank the reviewer for the positive evaluation.

Reviewer #2 (Remarks to the Author):

Revised manuscript NCOMMS-24-63684A titled "A bifunctional snoRNA with separable activities in guiding rRNA 2'-O-methylation and scaffolding gametogenesis effectors" has been substantially improved and the authors have addressed the most important concerns of the three reviewers. Concerning my own comments (reviewer #2), the authors now show in the revised manuscript that Mei2 accumulation in snR107 mutants is not due to indirect ribosome synthesis or translation defects as inactivation of other snoRNAs or the ribosome biogenesis factor *cgr1* does not induce Mei2 accumulation. Following meiosis induction, expression of meiotic transcripts in *cgr1*Δ cells is comparable to that observed in wild type cells, indicating that defects in ribosome assembly and/or activity do not indirectly change meiotic gene expression. The authors further show in the revised manuscript that Mmi1 and Mei2 interact with other snoRNAs besides snR107, in agreement with previous reports. This result rationalizes the snR107-independent interaction of Mmi1 and Mei2 with Fib1. In my opinion, the manuscript is now suitable for publication in Nature Communications journal.

We acknowledge the reviewer for the positive assessment of our work.

Reviewer #3 (Remarks to the Author):

My concerns have been fully addressed in the revised manuscript through the response to reviewers as well as significant new work and revisions to the manuscript. I support its publication.

We thank the reviewer for supporting publication of our work.